# Floral Elegance Meets Medicinal Marvels: Traditional Uses, Phytochemistry, and Pharmacology of the Genus *Lagerstroemia* L.

**DOI:** 10.3390/plants13213016

**Published:** 2024-10-28

**Authors:** Ziwei Yue, Yan Xu, Ming Cai, Xiaohui Fan, Huitang Pan, Donglin Zhang, Qixiang Zhang

**Affiliations:** 1Beijing Key Laboratory of Ornamental Plants Germplasm Innovation & Molecular Breeding, National Engineering Research Center for Floriculture, School of Landscape Architecture, Beijing Forestry University, Beijing 100083, China; yueziwei@bjfu.edu.cn (Z.Y.); xuyan@bjfu.edu.cn (Y.X.); htpan@bjfu.edu.cn (H.P.); zqx@bjfu.edu.cn (Q.Z.); 2Luoyang Landscape and Greening Center, Luoyang 471000, China; fanxiaohui01@163.com; 3Department of Horticulture, University of Georgia, Athens, GA 30602, USA; donglin@uga.edu

**Keywords:** crape myrtle, medical plant, biological compounds, pharmacological effects, corosolic acid, antidiabetic

## Abstract

The genus *Lagerstroemia* L. (Lythraceae), known for its exquisite flowers and prolonged flowering period, is commonly employed in traditional medicinal systems across Asian countries, where it has always been consumed as tea or employed to address ailments such as diabetes, urinary disorders, coughs, fevers, inflammation, pain, and anesthesia. Its diverse uses may be attributed to its rich active ingredients. Currently, at least 364 biological compounds have been identified from *Lagerstroemia* extracts, encompassing various types such as terpenes, flavonoids, phenolic acids, alkaloids, and phenylpropanoids. Extensive in vitro and in vivo experiments have examined the pharmacological activities of different extracts, revealing their potential in various domains, including but not limited to antidiabetic, anti-obesity, antitumor, antimicrobial, antioxidant, anti-inflammatory, analgesic, and hepatoprotective effects. Additionally, 20 core components have been proven to be associated with antidiabetic and hypoglycemic effects of *Lagerstroemia*. Overall, *Lagerstroemia* exhibit substantial medicinal potential, and the alignment between its traditional applications and contemporary pharmacological findings present promising opportunities for further investigation, particularly in food and health products, drug development, herbal teas, and cosmetics. However, evidence-based pharmacological research has largely been confined to in vitro screening and animal model, lacking clinical trials and bioactive compound isolations. Consequently, future endeavors should adopt a more holistic approach.

## 1. Introduction

The genus *Lagerstroemia* L., commonly known as crape myrtle, is a member of Lythraceae and renowned for its large and attractive flowers [1]. These plants are cultivated as ornamental trees in warm climates around the world. Moreover, they have been extensively utilized in traditional herbal medicine across various Asian countries, notably China, India, the Philippines, and Malaysia [2]. These plants contain flavonoids, terpenoids, phenols, tannins, volatile oils, alkaloids, fatty acids, and so on [3,4,5,6,7,8], exhibiting medicinal properties such as hypoglycemic [3], anti-obesity [9], antitumor [10], antiviral, antibacterial and antifungal [10,11,12,13], antioxidant [14,15], anti-inflammatory [16], analgesic [17], and hepatoprotective effects [18], as well as antidiarrhea effects [19]. *Lagerstroemia speciosa* Pers., also known as ‘Banaba’, has been the most widely used medicinal species, being one of the ten herbal plants approved by the Philippine Department of Health through its ‘Traditional Health Program’ [2]. Extracts of *L. speciosa* leaves have been used for the treatment of diabetes in traditional medicine for a considerable period, with initial documentation dating back to a study in 1940 [20]. Additionally, countries in Southeast Asia, such as the Philippines, Malaysia, Indonesia, Thailand, and India, have been using *Lagerstroemia* plants for over 1500 years as herbal teas for the prevention and treatment of diabetes, edema, and ulcers [21,22]. In traditional Chinese medicinal history, *Lagerstroemia indica* L. plants, notably, their leaves, flowers, roots, and bark, have all been harnessed for therapeutic purposes. The earliest recorded medicinal use dated back to the Diannan Materia Medica, with additional mentions in regional flora and medicinal plant compilations [23,24].

Despite their widespread use in traditional medicine, only a limited number of species have been subjected to comprehensive investigation and research to elucidate their bioactive compositions and mechanism-based pharmacological activities. A rigorous analysis of the chemical composition, bioactivities, and pharmacological characteristics of these medicinal valuable *Lagerstroemia* species is crucial for their accurate application and the development of evidence-based medicine. Regrettably, comprehensive reviews covering various aspects of the *Lagerstroemia* genus are quite limited, with most of them focusing on the medical research of a single species. The gap is significant, as the potential therapeutic value of these species remains largely untapped. Therefore, the primary objectives of this review were to systematically summarize general information on the traditional uses, phytochemistry, and pharmacology of all *Lagerstroemia* species, identify existing research gaps, and delineate future research directions to promote the conservation and effective utilization of *Lagerstroemia* plants. We believe this is the first comprehensive review to fully explore the medicinal potential of the *Lagerstroemia* genus in both traditional and modern medicine. It fills a significant gap in prior research that centered on individual species and establishes the possibility of the genus *Lagerstroemia* being used as versatile medicinal resources.

## 2. Taxonomy and Botanical Profile of *Lagerstroemia* Species

The Lythraceae comprises 31 genera with approximately 620 species [25]. Among them, *Lagerstroemia* is the most economically significant and renowned genus, which includes approximately 60 species [26]. These plants, deciduous or evergreen, present themselves as shrubs or trees characterized by smooth bark and leaves arranged oppositely, suboppositely, or in clusters. Their bisexual flowers, boasting radial symmetry, are borne in terminal or axillary panicles, typically with six petals or corresponding to the lobes of the sepals. As fruits mature, the woody capsules split into 3–6 valves, unveiling numerous seeds, each adorned with a winged apex. Vegetative propagation, particularly through cuttings, is often employed to maintain the desirable traits [27,28].

The WFO Plant List (http://www.worldfloraonline.org, retrieved on 25 July 2024) listed a total of 143 plant names in this genus, among which 61 were accepted names and the remaining 82 were synonymous or unresolved. *Lagerstroemia* plants are predominantly distributed in Eastern and Southern Asia, Northern Australia, and parts of Oceania [26,29]. China, as the primary center of origin for *Lagerstroemia* plants, boasts a widespread distribution of germplasm resources (Figure 1). The northernmost distribution extend to Beijing (116.4° E, 39.9° N), while the densest concentration of *Lagerstroemia* resources has been observed in the southwestern region. There are 25 different species in China, including 19 native species (Figure 1) and 6 introduced species (including *L. speciosa*, *Lagerstroemia siamica* Gagnep., *Lagerstroemia turbinata* Koehne, *Lagerstroemia loudonii* Teijsm. & Binn., *Lagerstroemia cuspidata* (Wall. ex C.B.Clarke) Craib, and *Lagerstroemia fauriei* Koehne). A genetic analysis revealed that wild species from various provinces in China constituted one clade, while the remaining, including *L. indica* and *L. fauriei* (the primary two progenitors for breeding modern crape myrtle cultivars), and modern cultivars from the USA, France, Japan, and China, were classified into two clades [1].

However, owing to human misconduct, climate change, and the widespread utilization of *Lagerstroemia* species for multifarious applications, natural plant populations are declining, with certain species classified as threatened. For instance, five species were listed on the IUCN (International Union for Conservation of Nature) Red List (*IUCN Red List of Threatened Species* Version 2022-2, https://www.iucnredlist.org, accessed on 19 December 2023) and the *China Biodiversity Red List* (Version 2013-9, http://www.iplant.cn/rep/protlist/4, accessed on 19 December 2023) listed nine *Lagerstroemia* species as threatened. Hence, the conservation of *Lagerstroemia* species is of paramount importance.

Most *Lagerstroemia* species have become important ornamental flowering shrubs due to their large and beautiful flowers, as well as their long flowering period [30]. Certain species exhibit vigorous regrowth after pruning, contributing to their value for afforestation on limestone hills and serving as precious timber resources [31]. According to Volza’s global export data, a majority of *L. speciosa* extracts were exported to the United States, Indonesia, and Vietnam (https://www.volza.com/p/Lagerstroemia-speciosa-extract/import/, accessed on 25 December 2023), finding applications in pharmaceuticals, dietary supplements, beverages, and health products [32]. It is evident that *Lagerstroemia* plants are not only esteemed for their ornamental value but also hold substantial potential for medicinal use.

## 3. Traditional Uses of *Lagerstroemia*

The use of traditional plant-based remedies holds significant cultural importance worldwide, with virtually every country having its methods of utilizing natural plants for treating various ailments [33,34,35]. In recent years, the field of ethnobotany and traditional pharmacology has garnered substantial attention in modern medicine, often serving as the foundation for exploring new potential medicinal plants [36]. According to a comprehensive survey of the literature, eight species within the genus exhibit extensive ethnobotanical and ethnomedicinal uses, primarily within Asian countries such as China, India, Philippines, Indonesia, Thailand, Bangladesh, and Iran (Table 1). *Lagerstroemia* plants have diverse traditional applications, ranging from medicinal purposes to the production of tea, dyes, flavorings, and culinary ingredients, with their predominant use being as a component in herbal teas and health supplements.

In Indonesia, *L. loudonii* is employed to treat hypertension, diabetes, urinary stones, diarrhea, dysentery, and hematuria [8,37,38,56,57,58]. The therapeutic properties of *Lagerstroemia ovalifolia* Teijsm. & Binn. make it a popular choice to combat diarrhea, malaria, and skin diseases in the Java region of Indonesia [40,59,60]. Additionally, the bark of *Lagerstroemia floribunda* Jack could manage diarrhea in Thailand [37,61].

In India, *Lagerstroemia lanceolata* Wall. has exhibited versatility in managing asthma, diabetes, chronic bronchitis, common cold, coughs, and topical applications to treat mouth ulcers, with the seeds having an anesthetic effect [17]. *Lagerstroemia parviflora* Roxb. has emerged as a valuable remedy, addressing fevers, infections, persistent ulcers, gastrointestinal strictures, and syphilis. It is also recognized for its effectiveness in controlling coughs, asthma, and bronchitis, as well as its role in aiding lactation among tribal women. Moreover, *L. parviflora* serves as a black dye for tanning and dyeing foam and cotton thread, and for producing edible chewing gum [4,41,43,62,63,64,65].

Notably, *L. indica* and *L. subcostata* holds significant importance in traditional Chinese medicinal herb, valued for its heat-clearing and detoxifying properties. Its bark, leaves, and flowers are commonly used in poultices or infusions to address various health issues, including boils, hemorrhages, dysentery, pulmonary tuberculosis, hemoptysis, and leucorrhea [23,24,55].

Furthermore, *L. speciosa* has gained significant attention for its traditional medicinal applications among all the species, as nearly every part of this plant, including the leaves, bark, flowers, fruits, and roots, possess notable medicinal characteristics [19,45,47,66]. Of all the natural products used for diabetes treatment, *L. speciosa* was listed among the 170 medicinal plants registered by the Ministry of Public Health in Thailand [61] and was also one of the 69 herbal plants promoted by the Department of Health in the Philippines. It has become a popular health beverage in East Asia and the United States and is widely used in the Philippines, Taiwan, and Japan as a tea preparation [67]. Its leaf extracts have been developed into functional food products, such as emulsions, soft gel capsules, and hard capsules, used in candies, bread, pastries, beverages, and other food products [32,68,69]. Despite generating considerable annual revenue, the sales of these products are currently limited in China, presenting a highly promising market opportunity for further development.

## 4. Nutritional and Phytochemical Profile of *Lagerstroemia*

*Lagerstroemia* plants are rich in a diverse array of nutrients, encompassing sugars, proteins, fatty acids, as well as amino acids, minerals, and essential oils. In addition, they contain phytochemicals such as flavonoids, phenolic acids, terpenoids, alkaloids, phenylpropanoids, coumarins, and lignans. A total of 364 compounds (Appendix A) were identified from various parts of these plants through our literature search, including 81 terpenoids (3 sesquiterpenoids, 4 diterpenoids, 20 triterpenoids, 20 tetracyclic triterpenoids, 19 pentacyclic triterpenoids, 6 steroids, and 9 other types), 85 phenolic compounds (10 phenolic acids, 9 polyphenols, 26 flavonoids, 8 anthocyanins, 20 tannins, and 12 other types), 24 phenylpropanoids (2 coumarins, 7 lignans, and 15 other types), 13 alkaloids, 19 fatty acids, 18 amino acids, 9 acyclic hydrocarbons, 5 alkanes, and 110 other types. To clarify the biological activity characteristics of these substances, we searched for them in relevant databases and found that many compounds had antioxidant, antidiabetic, antitumor, antibacterial and other properties in in vivo and in vitro studies and clinical trials (Appendix A), some of which have been widely applied in health supplements, food, industry and medicine. For example, as a natural antioxidant, squalene is used in anti-aging cosmetics and skin care products, vaccine development, functional food additives, and other fields [70]. The global squalene market was valued at $160 million in 2022 and expected to reach around $220 million by 2029 (https://www.mordorintelligence.com/industry-reports/squalene-market, accessed on 13 October 2024). In addition, catechin, widely found in tea and some fruits, is often used in functional foods and beverages as a natural preservative and antioxidant. It is also used in medicine to treat diabetes, obesity and cancer [71], contributing significantly to the global market share (https://www.reportsanddata.com/report-detail/catechin-market, accessed on 13 October 2024). However, there are still a large number of compounds that have not been fully studied, and the knowledge about their medicinal value is still a blank that needs to be filled.

### 4.1. Nutritional Profile

A qualitative analysis by gas–liquid chromatography (GLC) revealed the presence of various sugars in *L. indica* leaves, with polysaccharides constituting 20% *w*/*w* of the total extract [72]. The identified sugars included mannose (46.58%), sorbitol (25.74%), glucose (16.15%), galactose (5.21%), mannitol (2.430%), xylose (1.921%), ribose (0.620%), arabinose (0.520%), fructose (0.426%), and mannitol (0.392%), of which mannose was an important component and had the property of not interfering with normal blood sugar regulation, indicating that the polysaccharide extract of *L. indica* had the potential to be a sugar substitute for diabetic patients [73]. With 22.53% protein, 37.25% carbohydrates, and 12.23% ash on a dry weight basis, the leaves also contained high levels of protein, carbohydrates, potassium (K), calcium (Ca), magnesium (Mn), phosphorus (P), and some active chemical components, such as alkaloids, tannins, and saponins [74].

The composition of *L. speciosa* leaf extract consisted of 5.3% water, 1.2% protein, 6.3% lipids, 73.1% carbohydrates, and 2.1% dietary fiber, while the crude fat content in leaves and fruits was 3.36% and 0.26%, respectively. Additionally, leaves contained 13.76% fiber, 11.24% protein, and 51.20% carbohydrates, with mineral composition including 0.456% Ca, 0.443% K, 0.206% sulfur (S), and 0.017% Mn, whereas fruits exhibited varying mineral content, with K being the highest at 60.848%, followed by Ca, P, S, Fe, zinc (Zn), rubidium (Rb), copper (Cu), and strontium (Sr). Furthermore, a heavy metal analysis of leaf extract indicated mercury at 0.626 ppm, arsenic at 2.02 ppm, lead at 1.16 ppm, and cadmium at 0.26 ppm, all within the World Health Organization (WHO) limits [75].

Oil extracted from *L. lanceolata* seeds using light petroleum ether revealed a composition rich in palmitic acid (15.2%), stearic acid (6.10%), palmitoleic acid (3.80%), oleic acid (42.2%), and linoleic acid (24.8%) [17]. Notably, the dried fruits of *L. speciosa* had an oil content of 16.048 g/kg, with 20 fatty acids identified, including 16 long-chain saturated fatty acids and 4 unsaturated ones [76]. The seeds boasted high levels (85.76%) of oleic acid, linoleic acid, and other unsaturated fatty acids [77], potentially positioning them as valuable sources for edible oils and supplements, which could be utilized in reducing the risk of heart attacks, arteriosclerosis, and cancer [78].

The essential oil extracted from *L. speciosa* fruits via steam distillation underwent a GC–MS analysis and showcased prominent hydrocarbons such as methyl cyclohexane (60.9%), methyl benzene (18.2%), and o-xylene (3.04%), constituting 82.14% of the total oil, despite a modest yield of 0.068% [79]. Additionally, *L. speciosa* flower oil presented a profile featuring α-terpinene (10.38%), β-terpinene (8.45%), laurin (6.76%), limonene (2.6%), and α-santalol (3.14%) [80]. Meanwhile, the essential oil from leaves and flowers of *L. indica* was predominantly composed of cis-pinane (42.84%), chlorpyrifos (26.49%), and triacetylglycerol (15.08%) [81]. These distinct chemical profiles, dominated by hydrocarbons and terpenes, suggest potential applications for essential oils from *Lagerstroemia* species in the fragrance and industrial sectors [82].

*L. speciosa* fruit contained albumin (3.31 g/100 g), globulin (1.43 g/100 g), prolamin (2.09 g/100 g), glutelin (1.54 g/100 g), as well as a total of 18 amino acids, including 8 essential amino acids. The protein content and total amino acid content in the dry kernel of that fruit were 8.54 g/100 g and 7.27 g/100 g, respectively [83].

The results of these studies suggest that *Lagerstroemia* plants may serve as an important source of micronutrients, carbohydrates, sugar substitutes, proteins, oils, essential amino acids, and fatty acids for humans and animals. Despite the availability of staple foods and large agricultural investments around the world, many regions are still inevitably facing food security and nutrition issues. Assessing the nutritional composition of *Lagerstroemia* plants that have not yet been developed as a source of nutrition may further discover the potential of these plants as supplements to alleviate food shortages during special periods.

### 4.2. Phytochemical Profile

The phytochemical constituents of *Lagerstroemia* plants play crucial roles in their adaptation, defense mechanisms, and competitive survival, while also holding significant nutritional, medicinal, and industrial value for humans [84,85]. The discovery of multiple effective compounds provides empirical evidence for the traditional medicinal value of *Lagerstroemia*. Pentacyclic triterpenes, widely distributed in these plants [66,86,87,88,89], exhibit various biological activities, including antidiabetic, lipid-lowering, anticancer, antiviral, cardioprotective, antioxidant, anti-inflammatory, antibacterial, blood pressure-lowering, and immune-regulating effects [90,91,92,93,94,95,96,97,98]. The total triterpenoid content in *L. speciosa* can reach up to 166 mg/g [49], with corosolic acid (CA) emerging as the most extensively studied and commercially valuable component [99,100].

Numerous studies have indicated that the content variation of CA in *L. speciosa* is primarily controlled by genetics, influenced by plant age, organ type, and climate, and may be affected by drying processes, harvest times, and storage conditions [21,101,102,103]. The highest CA content was found in the leaves (0.89%) compared to other plant parts [101]. An analysis of methanol extracts from dried *L. speciosa* leaves using HPLC revealed CA contents ranging from 0.0100% to 0.7496% [104]. The distribution of CA in 12 natural populations of *L. speciosa* in the southern Western Ghats of India varied significantly, ranging from 0.005% to 0.868%, with higher content observed in northern populations compared to southern ones, suggesting a predominant influence of genetic factors over soil conditions in determining the CA content variation [21]. The CA content was notably higher in the red (old) leaves of *L. speciosa*, followed by petals, green leaves, flower buds, roots, and seeds, with the red coloration attributed to cyanidin 3-O-glucoside and a strong correlation observed between the content of CA and that substance [86,105,106]. The CA content in mature leaves (0.012–0.062%) of in vitro propagated seedlings of *L. speciosa* exceeded that in young leaves (0.004–0.007%), with differential expression of upstream limiting genes in terpene biosynthesis being relatively higher in young leaves, while downstream cytochrome P450 hydroxylase, catalyzing the final step of CA synthesis, exhibited greater expression in mature leaves [86], confirming the traditional belief of Filipinos that red-leaf Banaba tea possessed more pronounced medicinal effects. Moreover, collecting red leaves is relatively easy, unaffected by the growth and ornamental value of *L. speciosa*, and allows for the recycling of waste, while its ability to thrive in relatively barren land has contributed to land reclamation and increased land use value [106].

In addition to CA, other identified pentacyclic triterpenes include lupeol, betulin, α-amyrin, β-amyrin and so on, each with biological activities. Lupeol functioned as a competitive inhibitor of trypsin and chymotrypsin [107], betulin was substantiated for its anti-inflammatory activity in diverse experimental models [108], and α-Amyrin and β-amyrin were patented for application in the cosmetics industry as protective agents for hair and skin [109].

It is noteworthy that *Lagerstroemia* plants contain a significant number of phenolic compounds. High levels of polyphenols (10.3%) were detected in *L. speciosa* leaf extracts [110], with the 40% methanol extract of dry leaves containing total phenolic, total flavonoid, and tannin contents of 159.93 ± 0.87 μg/mg gallic acid equivalent (GAE), 9.37 ± 0.73 μg/mg quercetin equivalent (QE), and 80.5 ± 0.19 μg/mg GAE, respectively [111]. Additionally, 25 phenolic compounds and 8 new flavonoids were isolated and identified from *L. indica* leaves [72], with a high total phenol content (TPC) observed in autumn-fallen leaves at 465.71 ± 5.64 mg/g GAE in 96% ethanol extract [112]. Moreover, extracts from *L. floribunda* and *L. speciosa* flowers exhibited TPC of 418.50 ± 39.69 and 636.74 ± 44.14 mg/g GAE, respectively [113].

Cyanidin 3-O-glucoside, a potent antioxidant, was extracted from *L. speciosa* leaves, while malvidin 3,5-di-O-glucoside and malvidin 3-O-glucoside were isolated from petals [106]. The primary components of *L. speciosa* flower pigments were identified as four anthocyanin glycosides: delphinidin-3-O-glucoside, cyanidin-3-O-glucoside, petunidin-3-O-glucoside, and malvidin-3-O-glucoside [114]. Additionally, it was found that food additives and preservatives such as salt, glucose, sucrose, starch, citric acid, and sodium benzoate had no adverse effects on *L. speciosa* flower pigments, indicating their safety for use in the food processing industry as natural pigments with therapeutic effects [115].

The high content of terpenoids, phenolics, and flavonoids in *Lagerstroemia* extracts suggests their broad potential applications in the fields of food and nutrition [116]. By leveraging their antidiabetic, antioxidant, anti-inflammatory, and cardiovascular protective properties, they could be developed into functional foods, natural additives, and preservatives, thereby enhancing the nutritional value and health benefits of related products.

## 5. Pharmacological Activity

Modern pharmacological studies have demonstrated the medicinal potential of *Lagerstroemia* plants in treating a wide range of ailments, encompassing diabetes, cancer, obesity, high cholesterol, inflammation, hyperuricemia, diarrhea, pain, and Alzheimer’s disease, with the subsequent paragraph offering an overview of their various potential medicinal uses, further detailed and summarized in Table 2 and Table 3.

### 5.1. Antidiabetic Effects

According to the ‘2020 Chinese Guidelines for the Prevention and Treatment of Type 2 Diabetes’, the latest data using WHO diagnostic criteria revealed that the prevalence of diabetes in China had reached 11.2%, with over 90% of the cases being Type 2 diabetes, which was globally the most prevalent disease and ranked as the seventh highest cause of mortality [56]. Elevated blood sugar levels, the hallmark of diabetes, can lead to various complications including blindness, stroke, and cardiovascular diseases [155]. Oral hypoglycemic drugs such as pioglitazone and metformin provide temporary alleviation of symptoms, whereas insulin therapy, although highly effective, may lead to resistance and side effects like weight gain and lactic acidosis [156]. However, *Lagerstroemia* plants have shown to effectively lower blood sugar levels with minimal side effects in both in vivo and in vitro experiments [69,140,145,149,157,158], making their hypoglycemic effect a widely studied characteristic.

In numerous studies, extracts from *L. speciosa* leaves have demonstrated significant blood glucose-lowering activity, attributed to the inhibition of gluconeogenesis and the promotion of glucose oxidation through the pentose phosphate pathway [143]. Administration of the extracts to mice led to increased expression of liver peroxisome proliferator-activated receptor alpha (PPAR-α) mRNA, lipoprotein lipase (LPL) mRNA, and adipose tissue PPAR-γ mRNA [144], accompanied by a decreased body weight, fasting blood glucose, tissue weights, serum biomarkers, and adipose levels, as well as elevated insulin levels [3]. Additionally, extracts from *L. indica* and *Lagerstroemia indica* f. alba demonstrated stronger inhibition of α-glucosidase than acarbose and effectively treated dipyridyl-induced diabetic mice [118]. Furthermore, the 96% ethanol extract of *L. loudonii* leaves and fruits exhibited inhibitory effects on α-glucosidase, with fruits being 7 times stronger than acarbose and leaves being 24 times weaker, suggesting the potential development of fruits as herbal preparations for lowering blood glucose levels [56]. These findings underscore the diabetes-preventive and therapeutic potential not only of *L. speciosa* but also of other *Lagerstroemia* species.

Traditionally, CA was considered the active component for glucose reduction in *Lagerstroemia* plants. However, multiple studies suggested that the hypoglycemic activity of *Lagerstroemia* plants resulted from the combined effects of various chemical components, including CA [140], ursolic acid [159], PGG (1,2,3,4,6-Penta-O-galloyl-β-D-glucose), lagerstroemin, flosin B, reginin A [160,161,162], asiatic acid [163], valoneaic acid dilactone [164], and others [89,165,166] (Figure 2).

The extracts of *L. speciosa* contain abundant tannins (40%), with six ellagitannins (lagerstroemin, flosin B, stachyurin, casuarinin, casuariin, and 2,3-(S)-hexahydroxydiphenoyl-α/β-D-glucose) and four methyl ellagic acid derivatives (3-O-methylellagic acid, 3,3′-di-O-methylellagic acid, 3,4,3′-tri-O-methylellagic acid, and 3,4,8,9,10-pentahydroxydibenzo[b,d]pyran-6-one) identified from leaves, which exhibit strong activity in promoting insulin-like glucose absorption, impeding adipocyte differentiation and inhibiting glucose transportation. Interestingly, the tested ellagic acid derivatives demonstrated inhibitory effects on glucose transport for the first time [3,166], suggesting that tannins may stimulate glucose uptake like insulin by activating glucose transporter 4 (GLUT4) [3,167]. Moreover, three tannins isolated from *L. speciosa* (lagerstroemin, flosin B, and reginin A) increased glucose absorption in rat adipocytes [160], while gallotannins such as tannic acid exhibited similar activity [165]. Lagerstroemin is believed to exert insulin-like effects through a mechanism different from insulin by possibly binding to the extracellular portion of insulin receptors in a way that induces insulin receptor activation [168].

However, PGG is considered the most effective ellagitannin, with comparisons to published data indicating that gallotannins (such as PGG) seem to be more effective than ellagitannins (such as lagerstroemin) in insulin receptor binding, glucose transport induction, and insulin receptor activation [162]. Alpha-PGG not only enhanced glucose transport but also suppressed preadipocyte differentiation into adipocytes, indicating its distinctiveness from other medications in its ability to reduce blood sugar levels without promoting adiposity, and suggesting its potential as a novel generation of oral antidiabetic small molecule insulin mimetics [161].

After separating CA from the extracts, pure CA failed to stimulate glucose transport in cells, but it did not negate the possibility of CA to possess other antidiabetic activities, as it only eliminated the insulin-like glucose transport stimulation activity in adipocytes [165]. In addition to CA, oleanolic acid, asiatic acid, arjunolic acid, maslinic acid, and 23-hydroxyursolic acid were isolated from *L. speciosa* leaves [89,159], which also promoted glucose consumption in adipocytes and inhibited fat formation [169]. CA, ursolic acid, and betulinic acid exhibited good α-glucosidase inhibitory activity, with CA showing the strongest activity (IC_50_ = 3.53 μg/mL). CA, known as a plant-derived insulin, has a hypoglycemic mechanism that has been shown to induce the translocation of GLUT4, inhibit sucrose hydrolysis in the small intestine, regulate glucose metabolism, improve insulin resistance, regulate metabolism-related enzymes, and improve diabetic nephropathy [68]. Interestingly, CA inhibited the differentiation of 3T3-L1 adipocytes expression, downregulated PPAR-γ, and enhanced enhancer binding protein (C/EBP-α) mRNA expression, while also facilitating the uptake of [3H] glucose [170], suggesting its potential to lower blood glucose levels without promoting obesity, distinguishing it from many other antidiabetic medications. Due to its characteristics of lowering plasma glucose levels in in vitro, in vivo, and human studies, CA has become increasingly important in commercial and research fields [99,110,140] and has undergone three clinical pharmacological evaluations by the Food and Drug Administration (FDA) in the United States for managing and addressing type 2 diabetes and obesity [171].

Moreover, asiatic acid demonstrated antidiabetic effects by promoting glucose uptake in muscle cells [163], with some researchers believing it as a quality control compound with better hypoglycemic activity than CA within the *Lagerstroemia* plants [100]. Additionally, ursolic acid and ethyl ellagic acid demonstrated effective inhibition of Nuclear Factor-κB (NF-κB) activation and suppression of nitric oxide (NO) release, with ethyl ellagic acid showing significant inhibition of α-amylase and α-glucosidase activity [121]. Furthermore, CA, boswellic acid, tannic acid, and ursolic acid had inhibitory activity against the polyol enzyme aldose reductase in both laboratory and animal studies, suggesting *Lagerstroemia* plants containing such components could be used as protectants against diabetic complications [95,110,142]. In a word, the hypoglycemic activity of *Lagerstroemia* plants is a comprehensive effect produced by various chemical components through different pathways.

### 5.2. Anti-Obesity and Blood Lipid-Lowering Effects

The mice fed with a diet containing 5% hot-water extract of *L. speciosa* leaves showed increased body weight, significantly reduced weight of the perimetrium adipose tissue, and unaffected blood glucose levels, yet the hemoglobin A_1C_ was inhibited. Additionally, blood lipids remained unchanged, while the total lipid content in the liver decreased significantly (65% of the control group), primarily due to the reduction in triglyceride accumulation [9]. Moreover, oral *L. speciosa* seed oil (10 mg/kg, 20 mg/kg, 30 mg/kg) could significantly lower serum triglyceride, total cholesterol, and atherosclerosis index levels in hyperlipidemic mice, highlighting its notable lipid-lowering effects [172]. *L. parviflora* demonstrated significant dose-dependent effects (200 and 300 mg/kg b.w.) in alleviating obesity and hyperlipidemia in mice fed a high-fat diet, with no toxicity observed even at doses up to 3000 mg/kg, laying groundwork for anti-obesity drug development [149].

### 5.3. Antitumor Effects

Recent studies have demonstrated that CA and its structurally similar compounds (oleanolic acid, asiatic acid, maslinic acid, ursolic acid, betulinic acid) extracted from *Lagerstroemia* plants have therapeutic effects on various types of cancer [91], such as colorectal cancer [90], prostate cancer [93], colon cancer [98], gastric cancer [173], renal carcinoma [174]. Triterpenoid derivatives, betulinic acid, and 3β-acetoxyolean-12-en-28-acid from *L. indica* showed notable cytotoxicity against four human tumor cell lines (non-small cell lung adenocarcinoma, ovarian cancer cells, melanoma, and colon cancer cells) with IC_50_ values ranging between 3.38 and 6.29 μM [175]. Additionally, four phytosterol derivatives identified from *L. speciosa* seeds were recognized as nontoxic, noncarcinogenic, and non-mutagenic, making them potential candidates against breast cancer [176]. Plant sterols are renowned for their preventive effects on chronic conditions like diabetes, cardiovascular diseases, and cancer [171].

Numerous studies have shown that different solvent extracts of *L. speciosa* display promising anticancer properties. The ethanol extract induced considerable and concentration-dependent cytotoxicity effects and oxidative stress in human hepatocellular carcinoma (HepG2) cells, possibly attributed to the induction of oxidative stress and apoptosis through intrinsic or mitochondrial pathways [130]. When the ethanol extract was administered at a dose of 250 mg/kg in Benzo(a)pyrene [B(a)P]-induced lung tumor mice, it effectively alleviated various abnormal indicators, achieving notable therapeutic effects [132]. Similarly, the acetone extract, with a high content of gallic acid (49.712 mg/g) exhibited significant toxicity against cancer cells [131]. Furthermore, oxidized zirconium nanoparticles synthesized using *L. speciosa* leaves showed remarkable cytotoxicity against the breast cancer cell line Michigan Cancer Foundation-7 (MCF-7) [177,178,179], with the acetone extract demonstrating inhibitory effects of 92.9% and 87.13% at higher concentrations of 100 and 200 μg/mL, respectively. Additionally, the flower essential oil of *L. speciosa* at a dose of 50 μL/mL demonstrated cytotoxic effects of 13.33% and 31% against Dalton’s lymphoma ascites cells (DLA) and Ehrlich ascites carcinoma cells (EAC), respectively [80], collectively indicating the potential biomedical applications of *Lagerstroemia* plants extracts. However, most of the current research remains focused on in vitro cell experiments, with only a few studies having been validated in animal models.

### 5.4. Antiviral Effects

More than half of common cold cases are caused by Human rhinovirus (HRV), leading to billions of dollars in healthcare expenses annually. Ellagic acid and quercetin 7-glucoside in *L. speciosa* leaves exhibited higher inhibitory activity against HRV than ribavirin, acting by early inhibition of virus replication [11,133], while methanol extract from *L. speciosa* leaves, particularly orobol 7-O-D-glucoside (O7G), demonstrated broad-spectrum antiviral activity against various HRV types, suggesting potential for effective antiviral drugs against HRV [134].

In vitro studies evaluating ellagic acid and tannin from *L. speciosa* leaves and stems showed dose-dependent inhibitory effects on HIV-1 infection in TZM-bl and CEMGFP cell lines, with IC_50_ values ranging from 1 to 25 μg/mL. Ellagic acid inhibited reverse transcriptase, while tannic acid inhibited HIV-1 protease activity, explaining their potential as local anti-HRV drugs [135].

### 5.5. Antioxidant Effects

Recent research has confirmed that oxidative stress, induced by free radicals and oxidants, is linked to various diseases. Due to concerns over the carcinogenicity of synthetic antioxidants like BHT (Butylated Hydroxytoluene), BHA (Butylated Hydroxyanisole), and gallates, interest in natural antioxidant sources, particularly those rich in flavonoids and polyphenols, has surged [180]. Plants of the *Lagerstroemia* genus, distinguishing them for their antioxidant potency, boast an abundance of antioxidants, particularly flavonoids and polyphenols.

When investigating the antioxidant properties of various traditional herbal tea, it was found that *L. speciosa* tea exhibited superior antioxidant properties compared to green tea, surpassing oolong tea, black tea, and moringa (*Moringa oleifera* L.) tea, with the highest free-radical scavenging capacity among all teas [14,181]. In addition, the antioxidant properties of *L. speciosa* tea, particularly when subjected to freeze-drying during the drying process, were significantly enhanced, possibly attributed to ellagitannins such as lagerstroemin, flosin, and reginin [182,183]. Furthermore, the choice of extraction solvent influenced the antioxidant activity of *L. speciosa* leaves, with ethyl acetate and ethanol extracts showing superior activity compared to methanol and water extracts [129].

*L. parviflora*, abundant in phytochemicals such as polyphenols and flavonoids, exhibited significant DPPH (2,2-Diphenyl-1-picrylhydrazyl) radical scavenging ability in its ethanol extract, surpassing even that of ascorbic acid [124,184], with dried leaves yielding a flavonoid content of 66.9 mg/g. Interestingly, adding *L. speciosa* polyphenols to sausages at 0.02% concentration enhanced antioxidant performance without affecting sensory quality [185], indicating the potential of *Lagerstroemia* species as sources for functional foods and medicinal products [4,62].

Flowers of *L. speciosa* showed strong antioxidant effects, with higher phenolic content, free-radical scavenging capacity, and ferric reducing power compared to leaves. [15,52,151]. The antioxidant potential of *L. speciosa* and *L. floribunda* flower extracts was equivalent to ascorbic acid, with *L. speciosa* flower extracts exhibiting higher antioxidant characteristics, enzyme inhibitory activity, and cell protection due to their increased levels of major compounds like ellagic acid, epicatechin gallate, and quercetin. These flower extracts also inhibited MMPs (matrix metalloproteinases), crucial in preventing collagen degradation [186], indicating *Lagerstroemia* flowers as a promising natural solution for skincare formulations with anti-aging properties [187].

### 5.6. Antimicrobial Effects

With the rise in antibiotic-resistant bacteria due to antibiotic overuse, exploring new antimicrobial agents is essential, and plant compounds like flavonoids, polyphenols, volatile oils, and polysaccharides have shown antimicrobial properties. Certain components within *L. speciosa* exhibited promising antibacterial and antifungal activities [125]. Both ethanol and water extracts of *L. speciosa* leaves exhibited inhibitory effects against Gram-positive bacteria and Gram-negative bacteria, with water extracts demonstrating the highest efficacy (the inhibition zones for *Staphylococcus aureus*, *Bacillus subtilis*, *Pseudomonas aeruginosa*, and *Escherichia coli* were 15 mm, 15 mm, 17 mm, and 17 mm, respectively) [127]. Additionally, methanol extracts of *L. speciosa* leaves showed a dose-dependent inhibition against oral isolates of *Streptococcus mutans* strains. The inhibition zone exhibited a range of 1.0–2.6 cm at a 50 mg/mL extract concentration, narrowed to 0.8–2.1 cm at 25 mg/mL, and further reduced to 0.0–1.9 cm at 10 mg/mL [48]. Furthermore, extracts from the flowers and fruits of *L. speciosa* also demonstrated antibacterial and antifungal activities [52,67].

*L. indica* leaf extracts at different maturity stages inhibited various bacteria, with young leaves being the most effective, particularly against *Bacillus cereus* (20.0 mm) and *Shigella dysenteriae* (20.6 mm) [10]. Moreover, the effects of *L. indica* fruit volatile oil, primarily attributed to its composition of various straight-chain aliphatic hydrocarbons, alcohols, terpenes, and phenolic substances, were observed against *S. aureus*, *E. coli*, *Listeria*, and *Aspergillus niger*, with notably potent activity against *S. aureus* [44]. Additionally, antibacterial substances in *L. indica* leaves were identified as 4-methoxy apigenin-8-C-β-D-glucopyranoside, which exhibited potent antibacterial activity against a range of bacteria, including *Candida albicans* (MLC (minimum lethal concentration): 32 μg/mL), *S. aureus* (MLC: 16 μg/mL), *Salmonella enteritidis* (MLC: 16 μg/mL), *E. coli* (MLC: 16 μg/mL), and *L. monocytogenes* (MLC: 16 μg/mL) [123].

The methanol extract of *Lagerstroemia tomentosa* C.Presl leaves exhibited relatively weak inhibitory effects against *Mycobacterium tuberculosis*, reaching a maximum inhibition rate of 38% at a dose of 40 μg/mL [13]. Conversely, the leaf extract of *L. parviflora* demonstrated more effective inhibition against bacteria, including *S. aureus* and *Salmonella bongori*, with less impact on fungi, specifically *A. niger* [124].

These findings coincide with its traditional medicinal uses, such as the treatment of diarrhea, oral ulcers, and itchy rashes, as these conditions are closely linked to microbial imbalances. As shown in Appendix A, many components in *Lagerstroemia* extracts have demonstrated significant inhibitory effects against various microorganisms, making them promising candidates for use as antimicrobial agents in addressing challenges like antibiotic resistance and the side effects of synthetic compounds.

### 5.7. Anti-Inflammatory and Analgesic Effects

Studies conducted both in vitro and in vivo have indicated that leaf extracts of *L. ovalifolia* possess anti-inflammatory properties, showing efficacy in reducing inflammation in RAW264.7 macrophages stimulated by lipopolysaccharides (LPS) [40]. Additionally, they could act as anti-inflammatory agents in mice with LPS-induced acute lung injury by downregulating the activation of the Mitogen-Activated Protein Kinase (MAPK) and NF-κB pathways [59]. Similarly, *L. indica* extracts showed anti-inflammatory properties in vitro and in asthma mouse models, making them candidates for herbal remedies for allergic conditions, including asthma [128]. Extracts of *L. lanceolata* seeds and leaves also demonstrated significant analgesic and anti-inflammatory effects [17]. Additionally, *L. indica* extract without polysaccharides (Extract A) displayed stronger analgesic activity than the extract with polysaccharides (Extract B) [72].

The anti-inflammatory activity of *L. speciosa* leaf extracts was assessed using acute inflammation induced by carrageenan and chronic paw edema caused by formalin, with ethyl acetate extracts exhibiting dose-dependent efficacy and greater potency compared to the ethanol extracts [129]. In addition, the analgesic activity of *L. speciosa* was evaluated in Swiss albino mice using the acetic acid-induced writhing test, with the methanol crude extract showing significant analgesic effects at doses of 200 and 400 mg/kg body weight, resulting in inhibition rates of 35.38% and 53.85%, respectively, compared to sodium diclofenac (70.77%) [19].

The analgesic activity of *L. speciosa* bark chloroform extract was assessed by employing an acetic acid-induced gastric pain model in Swiss albino mice, displaying the most significant writhing inhibition at the highest test dose (500 mg/kg b.w.), reaching 50.7%, which exceeded the inhibition observed with the reference drug aspirin (38.4%). These findings validated the effective analgesic activity of *L. speciosa* bark, consistent with its traditional use for pain relief in Bangladesh [51], which might be attributed to the influence of arjunolic acid on the cyclooxygenase pathway [188].

### 5.8. Hepatoprotective Effects

The ethyl acetate portion of *L. speciosa* bark extract, as determined through a GC–MS analysis, revealed a high content of phenolic compounds and flavonoids, showing potent antioxidant and liver-protective effects, which improved the histopathology of liver by enhancing liver cell structure, reducing inflammation, and mitigating vascular and cellular degeneration in rats with sodium glutamate-induced liver toxicity [18].

Leaf extracts of *L. speciosa* exhibited hepatoprotective and anti-liver fibrosis effects in rats induced by carbon tetrachloride (CCl_4_) [54,154], showing significant reduction in hepatic hydroxyproline content, various serum enzyme levels, and total bilirubin levels at an oral dose of 100 mg/kg body weight, and thus leading to an improvement in the disrupted liver structure due to CCl_4_. This hepatoprotective effect was attributed to the synergistic action of various components within the plant rather than any single factor [152].

The hepatoprotective effect of water–methanol extracts from *L. indica* leaves against liver damage induced by carbon tetrachloride in rats was also investigated, with both extract A and B significantly lowering the levels of aspartate aminotransferase (AST), alanine aminotransferase (ALT), and alkaline phosphatase (ALP) compared to those in the control group, and the hepatoprotective effect of extract A was comparable to that of silymarin [72].

### 5.9. Anti-Hyperuricemia Effects

Comparing the inhibitory effects of *L. speciosa* leaf water extract on xanthine oxidase (XOD) with green tea (*Camellia sinensis* (L.) Kuntze), rooibos tea (*Aspalathus linearis* (Burm.f.) R.Dahlgren), and tochu tea (*Eucommia ulmoides* Oliv.), it was found that *L. speciosa* extract exhibited the strongest potential for inhibiting XOD [138]. The primary active components responsible for XOD inhibition isolated from the water extract of *L. speciosa* leaves were identified as valoneic acid dilactone and ellagic acid, with the inhibition of valoneic acid dilactone of XOD exceeding that of allopurinol, a drug clinically used as an XOD inhibitor [137]. These findings highlight the potential applications of *L. speciosa* leaf water extract in traditional medicine for preventing and treating hyperuricemia.

### 5.10. Other Effects

In addition to the previously mentioned functions, the antipyretic effect of methanol extract from *L. parviflora* leaves was studied in rats, revealing a significant reduction in yeast-induced fever comparable to that of paracetamol. At 200 mg/kg and 300 mg/kg doses, a significant fever-reducing effect was observed within five hours after administration, and the body temperatures of mice tended to be normal [64]. Furthermore, extracts from *L. indica* leaves demonstrated effective antipyretic activity in hyperthermic mice compared to acetylsalicylic acid [72]. It is a blessing that the 80% ethanolic extract of *L. indica* has exhibited promising results as a potential medication for Alzheimer’s disease, which might be attributed to the high content of β-carotene (117 mg/g), given that numerous previous studies have proven that carotenoids play a vital role in preventing various neurodegenerative diseases [153]. Methylene chloride and methanol extract of *L. indica* displayed significant anticoagulant activity (79% and 85%, respectively) [136]. Moreover, root extracts of *L. speciosa* reduced bowel movement frequency by 32.75% and 51.72% at doses of 200 and 400 mg/kg, respectively, compared to loperamide (58.62%) [19]. Additionally, fruit extracts of *L. speciosa* could also significantly prolong the average incubation period and reduce the frequency of defecation [16]. Various extracts of *L. speciosa* exhibited potent diuretic effects, with the water extract showing the most pronounced effect [75,189]. Furthermore, the methanol extract of *L. parviflora* leaves, in comparison to the control group, showed significant antitussive activity in a sulfur dioxide-induced cough model, comparable to that of the standard antitussive agent codeine phosphate [65]. It is noteworthy that all these findings support the traditional medicinal uses of *Lagerstroemia* species for hemostasis, diarrhea, diuresis, and cough management.

## 6. The Toxicity of *Lagerstroemia* Plants

Based on our understanding, leaf extracts of *L. speciosa* have shown promising treatments for diabetes, tumors, and hyperlipidemia with minimal adverse effects [19,157,158,190], while water extracts of *L. indica* leaves have demonstrated oral safety with LD_50_ values exceeding 5 g/kg [72], and leaf extracts of *L. parviflora* administered at a dose of 3000 mg/kg body weight in rats have not exhibited any signs of toxicity or fatality [149]. Nevertheless, both the flowers and fruits of *L. speciosa* exhibited toxicity [16,126], and leaf extracts showed sedation and posture defects in acute toxicity tests in mice [54]. Moreover, the essential oil extracted from steam-distilled *L. speciosa* fruit powder had an LC_50_ value of 1.701 μg/mL, indicating high toxicity [79], which may indicate that the extracts contained compounds with potential antitumor, antibacterial, or insecticidal properties.

CA in *Lagerstroemia* plants, although generally nontoxic, could induce acute kidney injury in some cases [191], and supplements containing *L. speciosa* extracts at recommended doses (8-48 mg/day) showed no negative reactions on the human body. However, higher doses could lead to lowered blood sugar levels, headaches, dizziness, and fatigue [192].

What is more, γ-sitosterol, constituting a significant portion (14.70–34.44%) in the leaves of *L. indica*, *L. speciosa*, *Lagerstroemia villosa* Wall. ex Kurz, and *L. loudonii* showed antihyperglycemic effects and induced DNA damage at higher concentrations, indicating that its toxicity and dosage should be carefully considered [61]. In conclusion, while *Lagerstroemia* extracts are generally safe, adverse effects in some studies highlight the need for further research into their pharmacological and toxicological effects.

## 7. Discussion

According to the WHO, plants are the best source for obtaining various medicines. In developing nations, where around two-thirds of the population lacked access to modern medical care, rising to 80% in Africa, a considerable segment of the populace continues to depend on traditional medicine and herbal remedies as their primary source of healthcare [184]. Historically relied upon for managing infections and illnesses, herbal plants were often prepared as infusions or teas by steeping dried plant parts in boiling water [181], but today, they are increasingly used as dietary supplements to combat prevalent illnesses like cancer, cardiovascular diseases, and depression [149], contributing to the steadily growing global herbal medicine market, valued at over $60 billion annually, with approximately 25% of modern pharmaceuticals derived from plant sources [79,149].

Currently, common applications of *L. speciosa* products include tea bags, dried herbs, ready-to-drink preparations, juice, beverages, as well as capsules and tablets for dietary supplements [68,99,193]. Similarly, other *Lagerstroemia* species hold the potential to be valuable sources of natural medicines and various products as renewable biological resources. After systematic data collation (Appendix A), it was found that the components such as corosolic acid, asiatic acid, arjunolic acid, betulinic acid, and oleanolic acid, known for their hypoglycemic activity in *L. speciosa*, were also identified in other species (*L. indica*, *L. calyculata*, *L. crispa*, *L. floribunda*, *L. limii*, *L. ovalifolia*, etc.) [2,56,100,194,195]. Although this study reviewed 20 components of *Lagerstroemia* plants that were previously studied for their hypoglycemic activity, a comprehensive database search in Appendix A revealed that there could be many more. Compounds such as 24-methylenecycloartanol, phytol, squalene, daucosterol, α-amyrin, arjunic acid, friedelin, lupeol, sitosterol, epicatechin, cinnamic acid, ferulic acid, ellagic acid, gallic acid, luteolin, rutin, 3-O-caffeoylquinic acid, and caffeic acid have all demonstrated antidiabetic effects in in vitro, in vivo, or clinical studies. Unfortunately, the concentrations of these ingredients in *Lagerstroemia* plants are still unknown, and it is not clear whether they play a major therapeutic role in the extracts. Furthermore, although the various compounds in Appendix A may have been shown to have active antioxidant, anti-inflammatory, anticancer, neuroprotection, antiviral, and liver protection effects, there is still a lack of systematic research on the correspondence between the pharmacological effects of *Lagerstroemia* plants and these components.

It is well known that with the advancement of modern medicine, plant-synthesized nanoparticles have potential biological applications. Zirconium oxide nanoparticles synthesized using water extracts of *L. speciosa* leaves exhibited excellent photocatalytic activity against methylene orange and azo dye, with degradation rates reaching as high as 94.58%, highlighting the potential of *L. speciosa* in environmental remediation [177,178,179]. Zinc oxide nanoparticles synthesized from *L. indica* demonstrated potential anticancer, antibacterial, and hemolytic activities, with minimum inhibitory concentrations for bacterial growth inhibition determined to be 88 μg/mL for *S. aureus*, 52 μg/mL for *E. coli*, 79 μg/mL for *P. aeruginosa*, and 72 μg/mL for *Klebsiella pneumonia* [12]. Analogously, silver nanoparticles (AgNPs) synthesized from *L. speciosa* leaf extracts (100 μL) exhibited substantial inhibitory effects on *K. pneumonia*, *S. aureus*, *E. coli*, and *P. aeruginosa* (*p* < 0.5) [196]. Furthermore, the formulation of self-micro-emulsifying systems using *L. speciosa* leaf extracts had the potential to enhance antidiabetic efficacy by approximately twofold [141] and improve hepatoprotective properties [152].

Based on the compiled components in Appendix A, it is evident that the identification of compounds is still at an early stage. For example, Bai et al. identified seven ellagitannins [166], Hou et al. isolated six pentacyclic triterpenoids from the leaves of *L. speciosa* [92], and three alkaloids [100] along with 25 compounds [72] were identified from *L. indica*. The low identification efficiency has hindered the development of its bioactive components and health benefits. To overcome this limitation, metabolomics approaches could be employed, utilizing modern analytical techniques to explore more extracts [197], such as ultra-performance liquid chromatography–mass spectrometry (UPLC–MS), gas chromatography–mass spectrometry (GC–MS), or ultra-high performance liquid chromatography–mass spectrometry/mass spectrometry (UHPLC–MS/MS).

To maximize the efficient utilization of plant materials, the extraction of *Lagerstroemia* species constituents requires ongoing exploration and experimentation. Conventional extraction methods such as Soxhlet extraction, maceration, and percolation have been reported [102,198,199,200,201], alongside advanced techniques such as microwave-assisted extraction, ultrasound-assisted extraction, supercritical fluid extraction, and enzyme-assisted extraction, which have been increasingly employed in recent years [112,201]. When comparing three-phase partitioning (TPP), batch extraction, and Soxhlet extraction, it was found that TPP was a more effective, simpler, and greener technique to maximize the extraction of CA from *L. speciosa* [102]. Moreover, an HPTLC method for the rapid and straightforward quantification of CA in *L. speciosa* was established [101]. As far as we know, transporting and storing plant extracts in powder form could significantly reduce production costs, with spray drying, using a boiling ratio of 1:7 or 1:3 (weight ratio) for *L. speciosa* extracts, achieving the highest drying capacity standards at 13,000 rpm and 120 °C, thereby holding significant implications for industrial applications [193].

To efficiently produce cost-effective CA and its bioactive derivatives for diabetes treatment on a global scale, an international patent has been granted for accumulating CA in plant cell cultures, with suspended *L. speciosa* cell cultures exhibiting a 56-fold increase in CA yield compared to naturally occurring leaf cell cultures [202]. Additionally, a rapid in vitro propagation method was developed for mass planting of *L. speciosa* using nodal explants in Schenk and Hildebrandt media [86].

In summary, research should not be limited to the pharmacological components of *Lagerstroemia* species and in vitro or in vivo experiments alone but also focus on employing advanced technologies, such as the use of synthetic nanoparticles, to optimize herbal formulations and enhance medicinal properties. Additionally, improving identification methods, extraction techniques, utilizing bioengineering to increase yield, and reducing extraction costs would help transform the bioactive ingredients of *Lagerstroemia* species into practical drugs and health products, ultimately benefiting human health.

## 8. Method

The paper extensively drew upon the diverse scientific literature, encompassing databases like Google Scholar, Web of Science, PubMed, Springer, and China National Knowledge Infrastructure (CNKI), along with published books and conference records. A systematic search was conducted using the following keywords. either individually or in combination to retrieve literature data spanning from 1940 to 2024: *Lagerstroemia*, crape myrtle, Banaba, medicinal, chemical constituent, traditional use, corosolic acid, ethnobotany, phytochemistry, pharmacology, antidiabetic, anticancer, anti-inflammation, anti-microbe, antioxidation. After screening titles and abstracts, original research articles and reviews focusing on the pharmacological activities, chemical constituents, or traditional medicinal uses of *Lagerstroemia* species were included. Studies unrelated to the topic, lacking experimental data, or using inadequate research methods were excluded, resulting in an initial collection of 442 relevant articles. Following a detailed reading of the full texts, 223 articles were retained for in-depth analysis and data extraction. Data were organized using Excel, with the traditional uses, identified compounds, and pharmacological activities of *Lagerstroemia* species categorized. Botanical names were verified using the WFO (World Flora Online) Plant List (http://www.worldfloraonline.org, accessed on 25 September 2024). Molecular structures were drawn using ChemDraw 22.0. The in vitro, in vivo functions. and clinical trials of the compounds listed in Appendix A were collected from the PubChem (https://pubchem.ncbi.nlm.nih.gov/, accessed on 11 October 2024), DrugBank (https://go.drugbank.com/, accessed on 11 October 2024), and ClinicalTrials.gov (https://clinicaltrials.gov/, accessed on 11 October 2024) databases.

## 9. Conclusions and Future Directions

This article delved into the traditional uses, phytochemistry, and evidence-based pharmacological properties of the genus *Lagerstroemia*. By analyzing the literature data, we found that only 8 of the nearly 60 species of *Lagerstroemia* had relevant records in traditional applications. A total of 364 compounds have been identified from the extracts of *Lagerstroemia* species, including terpenoids, phenolic acids, flavonoids, alkaloids, sterols and other types, some of which have been validated in in vitro, in vivo experiments and clinical trials. In modern pharmacological studies, it has been shown that the extracts of these plants have multiple functions such as antidiabetic, anti-obesity, antitumor, liver protection, antibacterial, and antioxidant effects. In addition, we summarized 20 components closely related to the hypoglycemic effects of *L. speciosa*.

In conclusion, the genus *Lagerstroemia* not only serves as an important ornamental and economically valuable woody plant but also demonstrates significant potential medicinal value in the field of healthcare. Isolating its metabolites may offer natural alternatives for pharmaceuticals, opening new commercial avenues for the industry. However, research on this genus is insufficient. Our comprehensive literature surveys revealed that, apart from *L. speciosa*, studies conducted on the chemical composition and pharmacological activities of most *Lagerstroemia* plants remained relatively scarce, and whether they have similar pharmacological activities to *L. speciosa* is still unclear. Moreover, evidence-based pharmacological activities were confined to in vitro bioactivity screening, with minimal exploration in vivo utilizing animal models, let alone the scarcity of human trials, resulting in a lack of precise exploration of the molecular mechanisms underlying their effects. In addition, there was also a lack of research on the isolation of bioactive compounds guided by bioassays.

Future research on medicinal plants of the *Lagerstroemia* genus should prioritize the exploration of traditional uses of different species across various ethnicities and geographical regions. Plant chemical experiments should be conducted to identify the active components present in the tissues of the genus *Lagerstroemia* and accurately qualify and quantify them, while also developing standardized extraction methods to enhance compound yields. Furthermore, scientific efforts should focus on bioassay-guided drug discovery based on traditional knowledge and contemporary research, involving mechanism-based in vitro and in vivo studies. To develop safe and effective formulations from *Lagerstroemia* species, clinical trials assessing pharmacokinetics, safe dosage ranges, and efficacy are also crucial. For endangered species listed in the IUCN Red List and China Biodiversity Red List, efforts should be made to determine the causes of population decline and raise awareness among local communities regarding their protection and sustainable utilization. Lastly, strategies should be devised to seamlessly integrate the significant ornamental and medicinal value of *Lagerstroemia* genus plants in landscaping practices, alongside the development of large-scale breeding and cultivation technologies to ensure resource sustainability.

## Figures and Tables

**Figure 1 plants-13-03016-f001:**
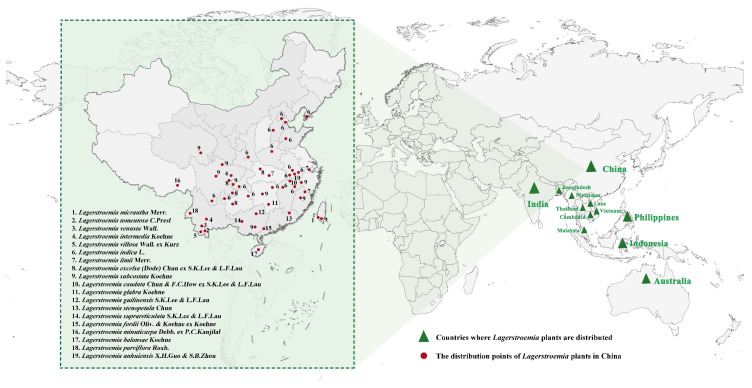
Distribution map of *Lagerstroemia* resources.

**Figure 2 plants-13-03016-f002:**
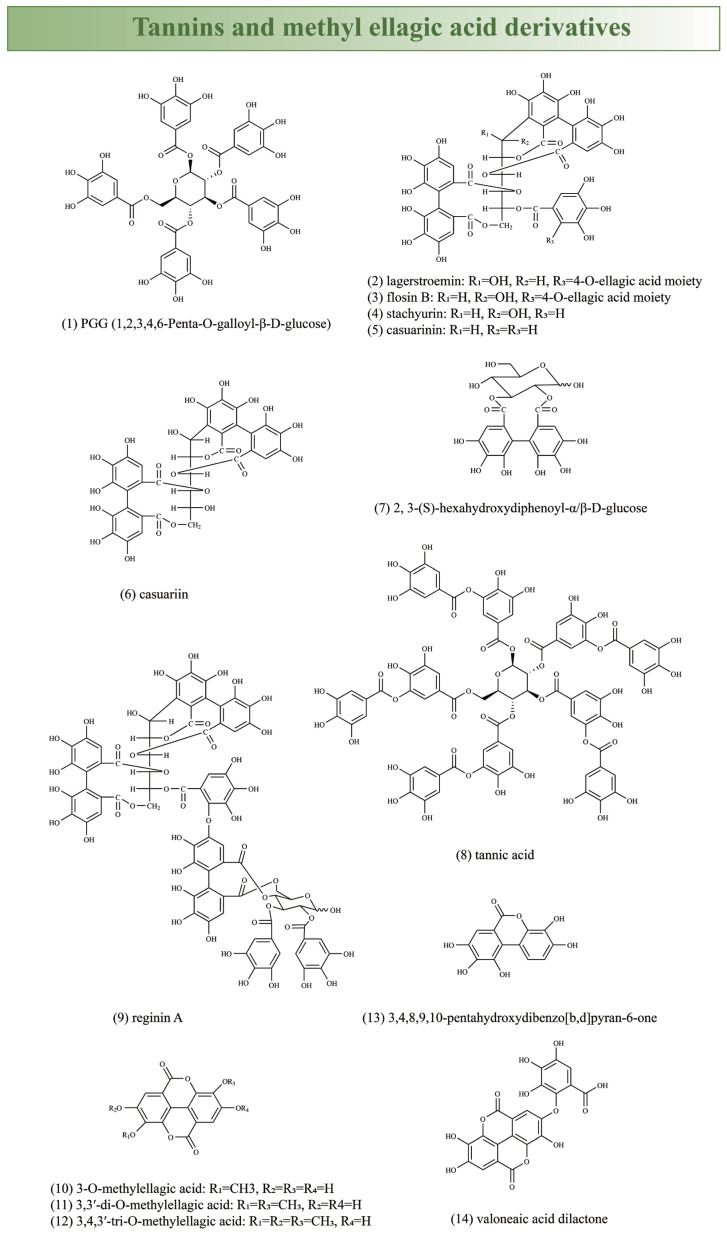
Potential hypoglycemic components (1–14 tannins and methyl ellagic acid derivatives, 15–20 pentacyclic triterpenes) in *Lagerstroemia* plants.

**Table 1 plants-13-03016-t001:** Traditional uses of *Lagerstroemia* spp. in different countries.

Traditional Uses	Preparation	Plant Part Used	Country	References
***Lagerstroemia floribunda* Jack**				
Treats diarrhea	-	Bark	Thailand	[37]
***Lagerstroemia indica* L.**				
Febrifuge, stimulant, and styptic	-	Stem bark	-	[10]
Diuretic and a drastic purgative	-	Bark, flower and leaf	-	
Treats cuts and wounds	Applied externally	Flower	-	
Astringent, detoxicant, and diuretic	-	Root	-	
Treats cold	Decoction	Flower	-	
Laxative and diuretic	-	Leaf, flower and bark	-	[6]
Treats asthma and hemostasis and as a detoxifier	-	-	-	
Clears heat and detoxifies, dispels wind and relieves itching; used for sore throat, ulcers, and itchy skin rashes	-	Bark	China	[24]
Clears heat and detoxifies, dispels wind and dampness, promotes blood circulation, and stops bleeding; used for sore throat, abnormal vaginal discharge, erysipelas, scabies, bruises, and internal or external bleeding from injuries	Decoction for washing; powdered for topical application.	Bark	China	[23]
Clears heat, detoxifies, cools blood, and stops bleeding; used for abscesses, swelling and toxins, breast abscesses, dysentery, eczema, and external bleeding from injuries	Decoction for washing; crush fresh leaves and apply topically	Leaf	China	
Clears heat, detoxifies, cools blood, and stops bleeding; used for leukorrhea, pulmonary tuberculosis with bloody cough, childhood convulsions, pediatric fetal toxins, sores, boils, carbuncles, and scabies	Decoction for washing; crush fresh flowers and apply topically	Flower	China	
***Lagerstroemia lanceolata* Wall.**				
Treats asthma, diabetes mellitus, chronic bronchitis, cold, cough, and local application for aphthae of the mouth	-	-	-	[17]
Narcotic	-	Seeds	-	
***Lagerstroemia loudonii* Teijsm. & Binn**				
Treats diarrhea	-	Bark	Thailand	[37]
Treats high blood pressure and diabetes	-	Seed	Indonesia	[38]
Treats urination stones, diabetes, and high blood pressure	-	Leaf	Indonesia	
Treats diarrhea, dysentery, and urinary blood	-	Bark	Indonesia	[39]
***Lagerstroemia ovalifolia* Teijsm. & Binn**				
Treats diarrhea	-	Bark	Indonesia	[40]
Malaria and dermatosis	-	Leaf	-	
***Lagerstroemia parviflora* Roxb.**				
Heals infections and persistent sores	-	Leaf	India	[41]
widely employed by tribal women to address lactation challenges	-	-	India	
Cures gastrointestinal strangulation and syphilis	-	Whole plant	India	
Manages coughs, fevers, asthma, and bronchitis	-	-	India	
Makes a black dye	-	Bark	India	
Produces edible sweet chewing gum	-	-	India	
Treat fever	Leaf juice	Leaf	India	[42]
Anti-asthmatic	-	Flower	India	[43]
Antitussive and astringent	-	-	India	[42]
***Lagerstroemia speciosa* L.**				
Treats malaria and foot fracture	Powder	Leaf	Malaysia	[44]
Decongestant, diuretic, and diabetes	Decoction	Leaf	Philippines, Thailand	[37,45]
Stimulant, antipyretic, alleviate abdominal discomfort	Decoction	Bark	Philippines	[45,46]
Anesthesia and pain relief of oral ulcer	Decoction	Seed, fruit	Philippines	[47,48]
Green leafy vegetables	-	Young leaves	Vietnam	[49]
ethnic medicines for lowering blood sugar	-	fruit and old leaf		
Purgative and facilitate bowel movements	-	Leaf, flower and bark	Philippines	[47]
Kidney and bladder inflammation, urinary issues, cholesterol reduction, hypertension, and diabetes management	Decoction or infusion	Leaf	Philippines	
Headaches, malaria, and cracked heels	Poultice, applied directly to the lesions	Leaf	Philippines	
Gastrointestinal upset, hematuria, stomach ache and depression	Decoction	Bark	Philippines	
Astringent, stimulant, febrifuge	-	Root	Philippines	
Diarrhea	Decoction	Bark	Malaysia	
Aphthous stomatitis	gargle with decoction	Fruit and root	Philippines	
Stomach problems	-	Root	Philippines	
Mouth ulcers	-	Root	-	[45,50]
Decrease blood glucose levels, promote weight loss	Herbal tea	Leaf	Philippines	[45]
Diabetes	Decoction; 50 g to a pint of boiling water, 4 to 6 cups daily	Leaf and dried Fruit	India, Bangladesh, Philippines	[16,45,51]
Toppings or ingredients in salads, soups, desserts, and drinks	-	Flower	-	[52]
Cleansing agent	-	Leaf	-	[50]
urinary tract infections	-	Leaf	-	
Prevents HIV infection	-	-	Bangladesh	[53]
Used by tribal peoples to treat heart disease	-	Leaf	-	[16]
Treats pain, purgative	Infusion	Bark	Bangladesh	[48,51]
Acute jaundice	-	Leaf	India	[54]
***Lagerstroemia subcostata* Koehne**				
Detoxify, dissipate blood stasis, and intercept malaria	Decoction	Flower, root	China	[55]
Carbuncle, snakebite, and malaria	Mash the fresh herb and apply directly to the lesions			

**Table 2 plants-13-03016-t002:** Pharmacological effects of *Lagerstroemia* spp. observed in in vitro studies.

Species	Part Used	Constituents/Preparations	Tested Pathogen/Cell	Results	Reference
**Hypoglycemic**					
*Lagerstroemia indica* L.	Stem and leaf	Ethyl acetate, chloroform, *n*-butyl alcohol, and water fraction of ethanol (95%) extract	α-Glucosidase and α-amylase	Inhibition rate of α-glucosidase: ethyl acetate fraction (73.60%), *n*-butanol alcohol fraction (59.78%), chloroform (55.26%); inhibition rate of α-amylase: chloroform fraction (61.46%)	[117]
*L. indica*	Flower	Petroleum ether, ethyl acetate, and *n*-butyl alcohol fraction of ethanol (70%) extract	α-Glucosidase	Showed highest inhibition rate in ethyl acetate fraction: IC_50_ = 4.45 μg/mL	[118]
*Lagerstroemia indica* Linn. f. *alba* (Nichols)	Flower	Petroleum ether, ethyl acetate, and *n*-butyl alcohol fraction of ethanol (70%) extract	α-Glucosidase	Showed highest inhibition rate in ethyl acetate fraction: IC_50_ = 4.09 μg/mL)	[118]
*Lagerstroemia loudonii* Teijsm. & Binn	Leaf and fruit	Ethanol (96%) extract	α-Glucosidase	Fruit extract: 7 times stronger inhibitor than reference drug Acarbose; leaf extract: 24 times weaker	[56]
*L. loudonii*	Stem bark	Ethanol (96%) extract	α-Glucosidase	IC_50_ = 79.479 ± 0.52 μg/mL	[57]
*L. loudonii*	Leaf	Ethanol (96%) extract	Dipeptidyl peptidase-IV	Inhibition rate 60.22 ± 2.01%	[58]
*Lagerstroemia speciosa* L.	Leaf	10% acetic acid in ethanol extract	α-Amylase	IC_50_ = 68.19 μg/mL	[5]
*L. speciosa*	Leaf	20% acetone/water extract	3T3-L1 fibroblasts	Inhibited adipocyte differentiation in 3T3-L1 cells	[3]
*L. speciosa*	Fruit	Ethanol (75%) extract	α-Glucosidase	IC_50_ = 121.26 ± 9.71 μM for compound 21	[119]
*L. speciosa*	Leaf and fruit	Water extract, ethanol (95%) extract	α-Glucosidase	IC_50_ = 4.29 μg/mL and 9.16 μg/mL for H_2_O extract, 2.64 μg/mL and 6.17 μg/mL for 95% EtOH extract of leaf and fruit	[29]
*L. speciosa*	Leaf	Water extract	α-Glucosidase	IC_50_ = 5.4 ± 0.5 μg/mL	[120]
*L. speciosa*	Leaf	Ethyl acetate extract (six pentacyclic triterpenes)	α-Glucosidase and α-amylase	Moderate inhibition of α-glucosidase (IC_50_ = 3.53 μg/mL) and weak inhibitory effect on α-amylase (IC_50_ = 100.23 μg/mL)	[89]
*L. speciosa*	Green and fallen leaf	Ethanol (80%) extract	α-Glucosidase and α-amylase	Inhibition rate of α-glucosidase: green leaf 29.06%, fallen leaf 30.49%; inhibition rate of α-amylase: green leaf 25.42%, fallen leaf 13.96%	[121]
**Antimicrobial**					
*L. indica*	Fruit	Low-polar solvent extract	*Staphylococcus aureus*, *Escherichia coli*, *Listeria monocytogenes*, *Penicillium glaucum*	Sensitivity to the antibacterial extract: *S. aureus* > *E. coli* > *L. monocytogenes* > *P. glaucum*	[44]
*L. indica*	Bark, leaf, and fruit	Petroleum ether, chloroform, methanol	Gram-positive bacteria (*S. aureus*, *Bacillus subtilis*), Gram-negative bacteria (*E. coli*, *Pseudomonas aeruginosa*), fungal strains (*Aspergillus oryzae* and *A. niger*)	Maximum antibacterial activity of petroleum ether extract of bark against *B. subtilis* (58.33 ± 0.88 mm); maximum antifungal activity of chloroform extract of bark against *A. niger* (40.33 ± 0.88 mm)	[122]
*L. indica*	Young, medium, and coarse leaf	Methanolic extract	*B. cereus*, *S. aureus*, *Proteus mirabilis*, *P. aeruginosa*, *Salmonella typhi*, *E. coli*, and *Shigella dysenteriae*	Halo range, minimum inhibitory concentration (MIC) and minimum bactericidal concentration (MBC): young leaf (10.2–20.6 mm, 100–145 mg/mL, 100–145 mg/mL); medium leaf (9.4–18.5 mm, 130–145 mg/mL, 130–160 mg/mL); coarse leaf (5.0–10.0 mm, 200–350 mg/mL, 145–300 mg/mL)	[10]
*L. indica*	Leaf	Methanol extract	4 pathogenic bacteria (*S. aureus*, *S. enteritides*, *E. coli*, and *L. monocytogenes*) and *Candida albicans*	MLC of compound ’4-methoxy apigenin-8-C-β-D-glucopyranoside: *C. albicans* (32 μg/mL), *S. aureus* (16 μg/mL), *S. enteritides* (16 μg/mL), *E. coli* (16 μg/mL), and *L. monocytogenes* (16 μg/mL)	[123]
*Lagerstroemia tomentosa* C.Presl	Leaf	Methanolic (70%) extract	*Mycobacterium tuberculosis*	Highest inhibition rate (38%)	[13]
*Lagerstroemia parviflora* Roxb.	Leaf	Methanol extract	2 bacteria (*S. aureus* and *S. bongori*), one fungus (*A. niger*)	Exhibited inhibitory activity against all tested microorganisms at different concentrations	[124]
*L. speciosa*	Fruit	CH_2_Cl_2_ extract	3 Gram-positive bacteria (*B. subtilis*, *B. cereus*, and *S. aureus*), 1 Gram-negative bacterium (*Klebsiella pneumonia*), 4 fungal strains (*Aspergilus flavus*, *A. niger*, *Rhizopus nigricans*, and *Fusarium equiseti*)	Showed poor antibacterial and antifungal properties in compound 1 while good in compound 2	[125]
*L. speciosa*	Flower	Methanolic extract	5 Gram-positive bacteria, 8 Gram-negative bacteria and 2 fungi	Largest inhibition zone against *S. aureus* (19.0 mm)	[126]
*L. speciosa*	Fruit (without seed)	Methanol and dichloromethane extract	4 bacterial and 3 fungi	MIC values: *S. aureus* (15–39 μg/mL), *E. coli* (16–38 μg/mL), *P. aeruginosa* (15–39 μg/mL), *B. subtilis* (14–39 μg/mL), *A. niger* (16–38 μg/mL), *A. flavus* (18–39 μg/mL), *C. albicans* (16–38 μg/mL)	[67]
*L. speciosa*	Flower	Methanol extract	4 bacteria (*S. aureus*, *B. cereus*, *Vibrio cholerae* and *E. coli*), 2 fungi (*C. albicans* and *Cryptococcus neoformans*)	Zone of inhibition: *S. aureus* (2.2 cm), *B. cereus* (1.9 cm), *E. coli* (1.7 cm), *V. cholerae* (1.7 cm), *C. albicans* (1.5 cm), *C. neoformans* (1.7 cm)	[52]
*L. speciosa*	Leaf	Methanol extract	*S. aureus*, *P. aeruginosa*, *S. typhimurium* and *E. coli*	Highest activity against *E. coli* (15 mm), lowest activity against *S. typhimurium* (7 mm)	[50]
*L. speciosa*	Leaf	Ethanol and water extract	*S. aureus*, *B. subtilis* (Gram-positive) and *Pseudomonas aeruginosa*, *E. coli*, (Gram-negative bacteria)	Zone of inhibition of ethanol and water extract: *S. aureus* (14 mm, 15 mm), *B. subtilis* (12 mm, 15 mm), *P. aeruginosa* (14 mm, 17 mm), *E. coli* (16 mm, 17 mm)	[127]
*L. speciosa*	Leaf	Methanol extract	12 oral isolates of *Streptococcus mutans*	Had a dose-dependent inhibition of cariogenic isolates but lower than that of standard antibiotic	[48]
**Anti-inflammation**					
*Lagerstroemia ovalifolia* Teijsm. & Binn	Leaf	Methanolic extract	Lipopolysaccharide (LPS)-stimulated RAW264.7 macrophages	Inhibited the production of NO, PGE2, interleukin (IL)-6, IL-1β, and tumor necrosis factor-α (TNF-α), suppressed the mRNA and protein expression of inducible nitric oxide synthase (iNOS), cyclooxygenase-2 (COX-2), inhibited the phosphorylation of MAPKs, reduced nuclear translocation of nuclear factor-κB (NF-κB)	[40,59]
*L. indica*	Whole plant	Ethanol (80%) extract	Jurkat cells, EoL-1 cells, and THP-1 cells	Strongly inhibited the protein release of IL-2, IL-4, IL-5, IL-13, and TNF-α in Jurkat cells	[128]
*L. speciosa*	Fruit	Ethanol (75%) extract	Macrophage BV-2 cell line	Anti-neuroinflammatory IC_50_ values of compounds 1 and 2: 8.29 ± 0.85 μM, 7.20 ± 1.55 μM	[119]
*L. speciosa*	Leaf	Ethyl acetate and ethanol extract	Human RBC membrane	30.01% and 37.50% HRBC protection in hypotonic solution of 50 μg/mL ethanol and ethyl acetate extract, respectively	[129]
**Anticancer**					
*L. indica*	Stem	Methanolic (80%) extract	Human tumor cell lines A549, SK-OV-3, SK-MEL-2, and HCT-15,	Showed potent cytotoxicity against tumor cell lines (IC_50_ = 3.3–6.29 μM)	[87]
*L. indica*	Stem and leaf	Ethanol (95%) extract	A2780, NCI-H1650, BGC-832, HepG2, HCT-116	Inhibitory effects: chloroform fraction on HepG2 (IC_50_ = 34.862 μg/mL), ethyl acetate fraction on HepG2 (IC_50_ = 43.261 μg/mL), and A2780 (IC_50_ = 46.673 μg/mL)	[117]
*L. speciosa*	Leaf	Extract contained 20% corosolic acid	HepG2 cells.	Caused significant cytotoxicity in HepG2 cells: membrane distortion and nuclear chromatin condensation; the loss of ΔΨm; interfere with Bax/Bcl-2 homeostasis; induce the pro-apoptotic marker genes (cytochrome c, Apaf-1, and caspases 9 and 3)	[130]
*L. speciosa*	Fruit	Ethanol (75%) extract	Tumor cell lines (HeLa, HepG2, and SGC-7901)	Showed obvious activity (IC_50_ = 15.90–39.13 μM) in compounds 13 and 18 when compared to the positive control cisplatin (IC_50_ = 3.31–16.6 μM)	[119]
*L. speciosa*	Leaf	Various solvents extract	MCF-7 cancer cell lines	Exhibited notable cytotoxicity against MCF-7 cell lines at 500 μg/mL	[131]
*L. speciosa*	Leaf	Ethanol (70%) extract	Human lung adenocarcinoma cells (A549)	Possessed cytotoxic activity against A549 cells: IC_50_ = 297.31 μg/mL (NRU assay), 41.23 μg/mL (MTT assay)	[132]
**anti-HRV**					
*L. speciosa*	Leaf	Methanol extract	HeLa cells and three rhinoviruses, HRV-2, -3, and -4	Toxicity levels for natural ellagic acid: 1.8, 2.3, and 2.2 times higher than ribavirin against HRV-2 (38 μg/mL), HRV-3 (31 μg/mL), and HRV-4 (29 μg/mL), respectively	[133]
*L. speciosa*	Leaf	Methanol extract (quercetin 7-glucoside)	HRV2 propagated in HeLa cells	Reduced the formation of a visible cytopathic effect, inhibited virus replication during the initial stage of infection, was more effective than ribavirin	[11]
*L. speciosa*	Leaf	Methanol extract	HRV species A (HRV1B, HRV2, HRV15, and HRV40), species B (HRV3, HRV6, and HRV14), pleconaril-resistant virus (HRV5)	Possessed broad-spectrum anti-HRVs activity: IC_50_ = 0.58–8.80 μg/mL, CC_50_ (50% cytotoxicity concentration) > 100 μg/mL	[134]
**Anti-HIV**					
*L. speciosa*	Stem and leaf	Aqueous and 50% ethanolic extract	HIV-1-infection in TZM-bl and CEMGFP cell lines	Showed a dose-dependent inhibition (IC_50_ = 1–25 μg/mL), inhibited reverse transcriptase and HIV protease	[135]
**Antithrombin activity**					
*L. indica*	-	Methylene chloride and methanol extract	Thrombin solution	Displayed 79% and 85%anticoagulant activity of methylene chloride and methanol extract, respectively	[136]
**Anti-hyperuricemia**					
*L. speciosa*	Leaf	Aqueous extracts	Xanthine oxidase (XOD)	A stronger XOD-inhibitory effect of valoneic acid dilactone than a clinical drug allopurinol, with a noncompetitive inhibition pattern for the enzyme with respect to xanthine	[137]
*L. speciosa*	Leaf	Aqueous extracts	XOD	Had a stronger potential in inhibiting XOD than *Aspalathus linearis* (Burm.f.) R.Dahlgren, *Camellia sinensis* (L.) Kuntze, and *Eucommia ulmoides* Oliv.	[138]

**Table 3 plants-13-03016-t003:** Pharmacological effects of *Lagerstroemia* spp. observed in in vivo studies (including animal and some human experiments).

Species	Part Used	Constituents/Preparations	Tested Subjects	Study Design	Results	Reference
**Hypoglycemic**						
*Lagerstroemia indica* L.	Leaf	Distilled water extract	Adult male albino rat	Daily oral dose of 100 mg/kg b.w. extract A or B for two months	Showed a decrease in serum glucose level after 4 and 8 weeks: extract A (22.5%, 44.9%), extract B (32.2%, 58.2%), reference drug metformin (46.2%, 66.4%)	[72]
*L. indica and Lagerstroemia indica* Linn. f. *alba* (Nichols)	Flower	Ethanol (70%) extract	Alloxan-induced diabetic mice	Oral administration of extracts at 125, 250, and 500 mg/kg/d for 7 days	Decreased fasting blood glucose, total cholesterol level, malondialdehyde content; increased superoxide dismutase level in serum	[118]
*Lagerstroemia balansae* Koehne	Leaf	Ethanol (95%) extract	Eight-week-old male B6.Cg-m +/+*Lepr^db^*/J (db/db) mice	Oral administration of extracts at 2 g/kg/d for 6 consecutive weeks	Decreased the blood-glucose and hemoglobin Alc; improve the glucose tolerance	[139]
*Lagerstroemia speciosa* L.	Leaf	Extracts, standardized to 1.13% corosolic acid	24 patients with metabolic syndrome	Single-center, randomized, double-blind, parallel, placebo-controlled; oral administration of extracts of 500 mg before breakfast and dinner for 12 weeks	Alleviated metabolic syndrome: significantly decreased systolic blood pressure (121.5 ± 12.9 vs. 116.3 ± 9.8 mmHg, *p* = 0.050), fasting plasma glucose (5.9 ± 0.4 vs. 5.7 ± 0.4 mmol/L, *p* = 0.034), triglycerides (2.3 ± 0.4 vs. 1.7 ± 0.5 mmol/L, *p* = 0.021), very low density lipoprotein (0.5 ± 0.1 vs.0.3 ± 0.1 mmol/L, *p* = 0.021), area under the curve of insulin (50,675 ± 14,309 vs. 37,983 ± 19,298 mmol/L, *p* = 0.017), and insulinogenic index (0.4 ± 0.2 vs. 0.3 ± 0.2, *p* = 0.047)	[140]
*L. speciosa*	Leaf	20% acetone/water extract	Alloxan induced diabetic mice	Oral administration of extracts at 0.25, 1.0, and 4.0 g/kg/d for 21 consecutive days	Lowered levels of body weight, fasting blood glucose, tissue weight, serum biomarkers, and body fat; increased final insulin levels	[3]
*L. speciosa*	Flower	Methanolic extract	Swiss-albino mice	Oral administration of methanolic crude extract at 200 mg/kg and 400 mg/kg doses	reduce blood sugar level by 48.85 and 56.12% at 200 and 400 mg/kg, respectively	[126]
*L. speciosa*	Leaf	Self-micro-emulsifying (SME) formulation of ethanol (50%) extract	Wistar albino male diabetic rats	Oral administration of SME formulation at 50, 100 mg/kg	Elevated the pharmacodynamic performance approximately twofold in SME; exhibited dose-dependent manner; comparable to the hypoglycemic effect of glimepiride; significantly decreased serum lipid	[141]
*L. speciosa*	-	Standardized extract	Wistar albino rats	Administration with corosolic acid, boswellic acid, ellagic acid, ursolic acid, and quercetin at a dose of 10 mg/kg body weight	Inhibited lens galactitol accumulation, with ursolic acid exhibiting the most potent effect	[142]
*L. speciosa*	Leaf	Hot-water extract	Male/female Albino rats	Oral administration of extracts to streptozotocin-induced 24 h fasted rats.	Showed prominent hypoglycemic activity; inhibited gluconeogenesis and promoted glucose oxidation via the pentose phosphate pathway	[143]
*L. speciosa*	Leaf	Capsules containing 10 mg of corosolic acid	31 subjects	Double-blind and cross-over, placebo-controlled; oral administration of corosolic acid of 10 mg	Had lower post challenge plasma glucose levels at 90 min than the placebo treatment subjects	[110]
*L. speciosa*	Leaf	Hot-water extract	4-week-old C57BL/KsJ-db/db, male mice	Oral administration of 0.5% water extract for 12 consecutive weeks	Reduced insulin, blood glucose, triglyceride and percent HbA1c; increased expressions of liver PPAR-α mRNA, adipose tissue PPAR-γ mRNA, and liver LPL mRNA	[144]
*L. speciosa*	Leaf	Soft gel capsule formulation and dry-powder-filled hard gelatin capsule formulation, both standardized to 1% corosolic acid (Glucosol^TM^)	56 type 2 diabetic volunteers	Randomized clinical trial; oral administration of a soft gelatin or hard gelatin formulation of 16, 32 and 48 mg Glucosol^TM^ for 15 days	Exhibited a superior percent reduction in blood glucose levels of soft gel formulation (4.9–30%) compared to dry-powder formulation (3.18–20.2%)	[145]
*L. speciosa*	Leaf	Banabamin, a tablet containing extract from Banaba tea	24 patients with mild type 2 diabetes	Crossover method; oral administration of 3 tablets or placebo t.i.d.	Significantly decreased blood glucose level	[69]
*L. speciosa*	-	Combined bioactive fraction of *Cinnamomum burmanni* (Nees & T.Nees) Blume and *L. speciosa*	patient with type 2 diabetes	Open and prospective clinical study; oral administration of fraction for 12 weeks	Improved glycemic control; enhanced insulin sensitivity, lipid profile, and adiponectin level; safe and tolerable	[146]
*L. speciosa*	Leaf	Hot-water extract	Male mice with type 2 diabetes	Oral administration of extract for 5 consecutive weeks	Regulated plasma glucose levels in noninsulin dependent diabetes mellitus	[147]
*L. speciosa*	Leaf	Water extract	Rabbits	oral administration of a decoction of leaves at 1 to 2 g/kg	Lowered the blood sugar with marked and prolonged effects in larger doses	[20]
**Anti-obesity**						
*L. speciosa*	Leaf	Hot-water extract	Female KK-A^y^ mice	Oral administration of 5% of a hot-water extract for 12 weeks	Significantly lowered body weight gain and parametrial adipose tissue weight; suppressed hemoglobin A1C; significantly decreased total hepatic lipid contents, attributed to decreased triglyceride accumulation	[9]
*L. speciosa*	Leaf	Capsule with Banaba leaf extract	56 participants (11 men and 45 women)	2 tablets of the Fat Conversion Inhibitor and 1 capsule of the Carbohydrate Absorption Inhibitor (contained leaf extract of Banaba) before each meal for 3 times per day	Reduced mean total body weight, percent body fat, and waist, hip, and chest circumference	[148]
*Lagerstroemia parviflora* Roxb.	Leaf	Methanolic extract	Male Wistar albino rat	Oral administration of extract at 200 or 300 mg/kg for 12 weeks	Significantly reduced total fat, fat percentage, blood glucose, insulin resistance, and lipid profile in a dose-dependent fashion	[149]
**Antitumor**						
*L. speciosa*	Leaf	Ethanol (70%) extract	Male Swiss albino mice weighing 22–25 g	Oral administration of extract at 250 mg/kg five days a week	Alleviated abnormal indicators in Benzo(a)pyrene [B(a)P]-induced lung tumor mice, such as weight, tumor-related enzymes, and genes	[132]
**Larvicide activity**						
*Lagerstroemia loudonii* Teijsm. & Binn	Leaf, bark, stem, and fruit	Ethanol (96%) extract	Phase III or IV instar *Aedes aegypti* mosquito larvae	Administration of extract at 250, 300, 350, 400, 450, and 500 μg/mL for 24 h	Showed larvicide activity in all organ extracts with the highest effect in fruit	[8]
**Anti-inflammation**						
*Lagerstroemia ovalifolia* Teijsm. & Binn	Leaf	Methanolic extract	Mice of lipopolysaccharide (LPS)-induced acute lung injury	Oral administration of extracts at 10 mg/kg, 20 mg/kg for 3 days	Suppressed inflammatory molecules and MAPK/NF-κB activation	[59]
*L. loudonii*	Seed	Purified light petroleum ether	Carrageenan-induced rat paw edema model	I.p. administration of extract at 50, 100 and 200 mg/kg	Inhibited the carrageenan induced rat paw oedema: 53.70%, 59.70%, 62.20% inhibition at the dose of 50, 100, and 200 mg/kg extracts, while 74.60% inhibition for standard drug diclofenac sodium (10 mg/kg i.p.)	[17]
*L. indica*	Stem and leaf	Ethanol (95%) extract	Rats	Administration of extract at 100 mg/kg, injected subcutaneously	Showed weak anti-inflammatory activity in ethyl acetate fraction (inhibition rate > 30%, *p* < 0.05)	[117]
*L. indica*	Whole plant	Ethanol (80%) extract	A mouse model of asthma	Oral injection administration of extract at 50 mg/kg, 250 mg/kg or 500 mg/kg between days 14 and 27, respectively	Significantly inhibited leukocytosis and eosinophilia in bronchoalveolar lavage (BAL) fluid and lung tissue samples; inhibited the increase in mucus secretion by goblet cells; blocked the production of reactive oxygen species in BAL fluid cells; blocked the protein expression of IL-5 in BAL fluid; weakly inhibited the concentration of ovalbumin-specific IgE in BAL fluid	[128]
*L. speciosa*	Leaf	Methanolic extract	Dextran sulfate sodium (DSS) induced ulcerative colitis in C57BL/6 mice	Oral administration of extracts at 100 and 200 mg/kg/d for 7 days	Significantly prevented DSS-induced inflammatory and ulcerative damages of the colon; reduced lipid peroxidation; restored the levels of innate antioxidants in the colon tissue	[150]
*L. speciosa*	Leaf	Ethyl acetate and ethanol extract	Female BALB/c mice	Oral administration of extract at 50 mg/kg, 2500 mg/kg	Ethyl acetate extract: had a significant dose-dependent anti-inflammatory effect against carrageen and formalin-induced paw edema in mice; ethanol extract: not in a dose-dependent manner and showed lesser activity in the formalin model	[129]
**Antioxidant**						
*L. indica*	Leaf	distilled water extract	Adult male albino rats	administration of extract A or B at 100 mg/kg	Percent of blood glutathione change: extract A (3.8%), extract B (6%), reference drug vitamin E (1.4%)	[72]
*L. speciosa*	Leaf	Ethanol extract	Rats	Oral administration of extract at 50 and 250 mg/kg	Decreased the level of tissue and serum malondialdehyde in a dose-dependent manner and increased the levels of superoxide dismutase, catalase, glutathione peroxidase, and glutathione	[54]
*L. speciosa*	Leaf	Standardized aqueous leaf extract having 1% corosolic acid fraction	Adult albino mice	Oral administration of extract at 50, 100, 150, 250, 500 mg/kg/day for 2 months	Duly reduced streptozotocin generated reactive intermediates and radical species; restored normal levels of antioxidative markers like superoxide dismutase, catalase, glutathione S-transferase, and reduce glutathione	[151]
**Hepatoprotective**						
*L. indica*	Leaf	Distilled water extract	Adult male albino rats	Oral administration of extract A or B at 100 mg/kg/d for 1 month	Notably decreased the liver enzymes AST, ALT, and ALP (extract A > B)	[72]
*L. speciosa*	Bark	Ethanol (70%) extract	Sprague Dawley adult male rats	Oral administration of extract at 100 and 200 mg/kg/day for 2 weeks	Enhanced liver histopathology by restoring hepatocellular architecture, reducing inflammation and mitigating vascular and cellular degeneration	[18]
*L. speciosa*	Leaf	Ethanol (50%) extract	Rats with induced liver toxicity by carbon tetrachloride	Oral administration of self-micro-emulsifying formulation of extract at 50 and 100 mg/kg every 72 h for 14 days.	Effectively protected serum enzymes; prevented the rise in levels of lipid peroxidation; increased the glutathione, superoxide dismutase and catalase contents; showed protection at the dose of 100 mg/kg comparable to normal control and standard	[152]
*L. speciosa*	Leaf	Ethanol extract	Male Wistar rats	Treated with extracts at dose of 50 and 250 mg/kg body weight for seven days	Significantly normalized serum and liver tissue parameters to near-normal levels	[54]
**Analgesic**						
*L. indica*	Leaf	Distilled water extract	Adult male albino rats of Sprauge Dawely Strain	Oral administration of extract A or B	Exhibited superior analgesic activity in extract A (93.8%) compared to extract B (74.6%)	[72]
*Lagerstroemia lanceolata* Wall.	Seed	Purified light petroleum ether extract	Mice with acetic acid-induced writhing method and hot-plate method	Oral administration of extract at 10, 20 and 40 mg/kg	Acetic acid-induced writhing test: demonstrated significant analgesic effects; achieved reductions of 28.82%, 48.58%, and 75.73% at doses of 10, 20, and 40 mg/kg; outperformed a standard drug diclofenac sodium at 5 mg/kg (63.77%); hot-plate test: significantly elevated pain thresholds	[17]
*L. speciosa*	Flower, root	Methanolic extract	Acetic acid induced writhing in Swiss albino mice	Oral administration of extract at 200 and 400 mg/kg	Produced 35.38% and 53.85% (*p* < 0.001) of writhing inhibition at 200 and 400 mg/kg, and 70.77% inhibition in standard diclofenac sodium; exhibited dose-dependent inhibition	[19,126]
*L. speciosa*	Bark	Chloroform extract	Acetic acid-induced gastric pain model in Swiss albino mice	Oral administration of extract at 250 and 500 mg/kg	Exhibited notable inhibition of writhing (50.7%) at 500 mg/kg	[51]
*L. speciosa*	Fruit	Ethanolic extracts	Acetic acid-induced writhing model in mice	Oral administration of extract at 250 and 500 mg/kg	Produced approximately 45.95% and 70.27% writhing inhibition at 250 and 500 mg/kg, respectively	[16]
**Anti-diarrhea**						
*L. speciosa*	Root	Methanolic extract	Castor oil-induced method in Swiss albino mice	Oral administration of extract at 200 and 400 mg/kg	Dose-dependently reduced diarrhea, comparable to a standard drug loperamide	[19]
*L. speciosa*	Fruit	Ethanolic extracts	Swiss albino mice	Oral administration of extract at 50 and 500 mg/kg	Delayed the onset of diarrheal episode and decreased the frequency of defecation	[16]
**Anti-Alzheimer**						
*L. indica*	Leave	Ethanol (80%) extract	Aluminum chloride-induced Alzheimer’s disease rats	Oral administration of extract at 500 mg/kg	Exhibited neuro-modulating effect in Al-induced neurotoxicity	[153]
**Antipyretic**						
*L. parviflora*	Leaf	Methanolic extract	Albino rats	Oral administration of extract at 100, 200, and 300 mg/kg after the yeast injection	Antipyretic effect comparable to a standard drug paracetamol	[64]
*L. indica*	Leaf	Distilled water extract	Female albino rats of Sprauge Dawely Strain	Oral administration of extract A or B at 100 mg/kg	Significantly decreased the temperature of the hyperthermal rats	[72]
**Antitussive**						
*L. parviflora*	Leaf	Methanolic extract	A cough model induced by sulfur dioxide gas in mice	Oral administration of extract at 100, 200, and 300 mg/kg	Exhibited a significant dose-dependent antitussive activity, comparable to a standard drug, codeine phosphate	[65]
**Diuretic activity**						
*L. speciosa*	Leaf	Ethyl acetate, ethanol, methanol, and water extract	Male Wistar rats	Oral administration of ethyl acetate, ethanol, or water extract at 250 mg/kg	Water extract: exhibited superior diuretic properties and increased urine excretion of Na^+^ and K^+^	[75]
**Anti-fibrotic effect**						
*L. speciosa*	Leaf	Ethanol extract	Male albino Wistar rats	Oral administration of extract at 100 mg/kg/d for 28 days	Reduced the hydroxyproline content of the liver, various serum enzymes level, and total bilirubin; improved the architecture of liver deranged by CCl_4_	[154]

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
