# Peer review of "Floral Elegance Meets Medicinal Marvels: Traditional Uses, Phytochemistry, and Pharmacology of the Genus Lagerstroemia L."

_plants, 2024, doi:10.3390/plants13213016_

Round 1
Reviewer 1 Report
Comments and Suggestions for Authors
The authors have carried out an important review about the Traditional Uses, Phytochemistry, and Pharmacology of Lagerstroemia L. My only suggestion to the authors would be to incorporate and cite in the text the following bibliographical references:
.-“Lagerstroemia Speciosa (L.) Pers Leaf Extract Attenuates Lung Tumorigenesis via Alleviating Oxidative Stress, Inflammation and Apoptosis” Amria M. Mousa et al. Biomolecules 2019, 9, 871; doi:10.3390/biom9120871 .-
"Preliminary results on the chemical components of Lagerstroemia indica L. in the conditions of southern Uzbekistan," M A Sharopova et al. https://doi.org/10.1051/e3sconf/2024 51003032 5 10 03032 ESDCA2024
Comments on the Quality of English LanguageThe syntaxis and English langauge along he manuscript is adequate.
Author Response
We sincerely appreciate the time and effort you have dedicated to reviewing our manuscript titled “Floral Elegance Meets Medicinal Marvels: Traditional Uses, Phytochemistry, and Pharmacology of the genus Lagerstroemia L.”. Please find the detailed responses below and the corresponding corrections highlighted in the re-submitted files.
Comment:
My only suggestion to the authors would be to incorporate and cite in the text the following bibliographical references:
“Lagerstroemia Speciosa (L.) Pers Leaf Extract Attenuates Lung Tumorigenesis via Alleviating Oxidative Stress, Inflammation and Apoptosis” Amria M. Mousa et al. Biomolecules 2019, 9, 871; doi:10.3390/biom9120871
“Preliminary results on the chemical components of Lagerstroemia indica L. in the conditions of southern Uzbekistan,” M A Sharopova et al. https://doi.org/10.1051/e3sconf/2024 51003032 5 10 03032 ESDCA2024
Response:
Thank you very much for your suggestion. I have carefully read the two references you recommended, and they are highly relevant to the research in the manuscript. I have appropriately cited them and added relevant content in the text. For the first reference, I summarized the in vitro and in vivo experimental entries in Tables 2 and 3, and cited it in the main text while ensuring smooth flow of the narrative. For the second reference, I have provided a summary and citation at the appropriate places in the main text.
The supplementary entry in Table 2:
|
L. speciosa |
leaf |
ethanol (70%) extract |
human lung adenocarcinoma cells (A549) |
possess cytotoxic activity against A549 cells: IC50 = 297.31 μg/mL (NRU assay), 41.23 μg/mL (MTT assay) |
[134] |
The supplementary entry in Table 3:
|
L. speciosa |
leaf |
ethanol (70%) extract |
male Swiss albino mice weighing 22–25 g |
oral administration of extract at 250 mg/kg five days a week |
alleviate abnormal indicators in Benzo(a)pyrene [B(a)P]-induced lung tumor mice, such as weight, tumor-related enzymes and genes |
[134] |
The citation of the first reference in the main text (lines 462-464):
Numerous studies had shown that different solvent extracts of L. speciosa displayed promising anticancer properties. The ethanol extract induced considerable and concentration-dependent cytotoxicity effects and oxidative stress in human hepatocellular carcinoma (HepG2) cells, possibly attributed to the induction of oxidative stress and apoptosis through intrinsic or mitochondrial pathways [132]. When the ethanol extract was administered at a dose of 250 mg/kg in Benzo(a)pyrene [B(a)P]-induced lung tumor mice, it effectively alleviated various abnormal indicators, achieving notable therapeutic effects [134]. Similarly, the acetone extract, with a high content of gallic acid (49.712 mg/g), exhibited significant toxicity against cancer cells [133].
The citation of the second reference in the main text (lines 251-253):
The essential oil extracted from L. speciosa fruits via steam distillation underwent GC-MS analysis, showcased prominent hydrocarbons such as methyl cyclohexane (60.9%), methyl benzene (18.2%), and o-xylene (3.04%), constituting 82.14% of the total oil, despite a modest yield of 0.068% [80]. Additionally, L. speciosa flower oil presented a profile featuring α-terpinene (10.38%), β-terpinene (8.45%), laurin (6.76%), limonene (2.6%), and α-santalol (3.14%) [81]. Meanwhile, The essential oil from leaves and flowers of L. indica was predominantly composed of cis-pinane (42.84%), chlorpyrifos (26.49%), and triacetylglycerol (15.08%) [82]. These distinct chemical profiles, dominated by hydrocarbons and terpenes, suggest potential applications for essential oils from Lagerstroemia species in the fragrance and industrial sectors [83].
Reviewer 2 Report
Comments and Suggestions for Authors
The manuscript presents a broad review of the genus Lagerstroemia and its traditional medicinal uses, phytochemistry, and pharmacological potential. However, there are significant shortcomings that need to be addressed. The paper suffers from a lack of depth, superficial data integration, and inadequate critical analysis. While the topic is of interest, the execution lacks precision, organization, and scientific rigor necessary for a high-impact journal.
* I think that the major weakness of this paper is that it largely recapitulates well-known information without providing any new insights or synthesizing the data in a meaningful way. For a review paper, there needs to be a more critical approach to summarizing existing knowledge rather than repeating previously established facts.
* The gap in research regarding Lagerstroemia genus is noted, but the problem could be highlighted more clearly in the introduction section.
* The methodology is section needs enhancement, as there is no clear description of literature research methodology, inclusion/exclusion criteria, or any systematic method employed to ensure the comprehensive selection of studies.
* The extensive lists of bioactive compounds, found in Lagerstroemia species, and their supposed effects are presented with minimal commentary on the strength of the evidence supporting these claims. Are these compounds genuinely responsible for the reported effects? Have they been tested in robust in vivo models or clinical trials? The discussion is shallow and lacks the necessary scientific skepticism. What I suggest is to add new columns with the model (whether in vitro/in vivo …)
* The inclusion of evolutionary and phylogenetic information feels disconnected from the main theme of the review. The brief discussion on the divergence between Lagerstroemia, and Punica granatum is irrelevant in the context of the phytochemistry and medicinal uses (lines 90-92), making the section seem forced. There’s little to no explanation of how these evolutionary details contribute to the understanding of Lagerstroemia’s medicinal potential (lines 88-93).
* Figure 1 is blurry and not quite readable; I suggest providing a better resolution figure.
* Tables and figures are cluttered with excessive detail that distracts from the main message. For example, Table S1, with 364 compounds, is excessive and not usable for the average reader. The data should be filtered to highlight key compounds or those with the most pharmacological relevance.
The paper is generally good, well-written (except in some subsections), I suggest revising the paper before its acceptance as this paper contributes as a full body of knowledge on Lagerstroemia species’ medicinal uses and phytochemistry, and paves the way for future research on these valuable plant species.

Reviewer 3 Report
Comments and Suggestions for Authors
The manuscript “plants-3250362” entitled “Floral Elegance Meets Medicinal Marvels: Traditional Uses, Phytochemistry, and Pharmacology of Lagerstroemia L.” by Yue et al. provides a comprehensive overview that aimed to systematically summarize general information on the traditional uses, phytochemistry, and pharmacology of all Lagerstroemia species, identify existing research gaps, and delineate future research directions to promote the conservation and effective utilization of Lagerstroemia plants. In addition, this paper provides a rigorous analysis of the chemical composition, bioactivities, and pharmacological characteristics of the medicinal valuable Lagerstroemia species. Moreover, it is the first review where an extensive number of Lagerstroemia species are subjected to comprehensive investigation and research to elucidate their bioactive compositions and mechanism-based pharmacological activities.
For publication in the “Plants” journal, the topic and content are appropriate. The subject of the review article is interesting and topical, with scientific and practical importance. The introduction follows the subject and is correctly presented. Scientific articles of recent date and in concordance with the topic of the study were consulted. The methodology of the study was clearly presented, and appropriate to the proposed objectives. The obtained results have been fully analyzed and the discussions are appropriate, in the context of the results. The editing and linguistic quality are good. In addition, it is easy to follow by the reader, the figures and tables give good summaries, and the text is edited to a thoughtful conclusion part. The scientific literature, to which the reporting was made, is recent and representative of the field covering the chemical composition, bioactivities, and pharmacological characteristics of the medicinal valuable Lagerstroemia species. However, some points need attention for the article to be published. I would like to recommend the publication of this article and a revision is required for the reasons listed below:
- Title of Manuscript: It is better to revise the title of the manuscript as follows: “Floral Elegance Meets Medicinal Marvels: Traditional Uses, Phytochemistry, and Pharmacology of the genus Lagerstroemia L.”
- Line 23: The correct is: ” The genus Lagerstroemia L. (Lythraceae), known for ….”
- Lines 108-109: Please refer to the 25 different species. Specifically, determine that the 19 native species are shown in Figure 1 and present the 6 introduced species.
- Figure 1: The quality of the figure (map) is very low. Please fix this problem.
- Authors should further check for typesetting errors in the text.
- Discussion: Based on the current discussion used in your text, please enhance this part by further referring to previous studies on this topic.
- Conclusions: Please enhance the analysis of your conclusions which should provide a good reflection on the results and novelty of this work, as well as the future prospects that you mentioned.
Finally, be consistent with the formatting of your manuscript. The reviewer recommends the authors carefully check and revise the manuscript according to the "Instructions for authors".
Thank you for your consideration.
Comments on the Quality of English LanguageThe English language in this manuscript is good. Authors should further check for typesetting errors.
Round 2
Reviewer 2 Report
Comments and Suggestions for Authors
All the comments were incorporated and the paper recommended for publication
Comments on the Quality of English LanguageMinor editing of English language required.
Author Response
Thank you very much for your feedback on our manuscript. We sincerely appreciate the time and effort you have dedicated to reviewing our work.
We have further reviewed the English language of the manuscript and revised it based on the editor's comments to ensure it meets the journal's requirements. Specific changes include the clear definition of the abbreviations CA (line 277) and MLC (page 11, Table 2, reference 10) when they first appear in the text.